# Environment Generation for Zero-Shot Compositional Reinforcement Learning

**Izzeddin Gur, Natasha Jaques, Yingjie Miao, Jongwook Choi,**
**Manoj Tiwari, Honglak Lee, Aleksandra Faust**
Google Research, Brain Team
Mountain View, California, 94043
{izzeddin, natashajaques, yingjiemiao, mjtiwari, sandrafaust}@google.com
{jwook, honglak}@umich.edu

## Abstract

Many real-world problems are compositional – solving them requires completing interdependent sub-tasks, either in series or in parallel, that can be represented as a dependency graph. Deep reinforcement learning (RL) agents often struggle to learn such complex tasks due to the long time horizons and sparse rewards. To address this problem, we present Compositional Design of Environments (CoDE), which trains a Generator agent to automatically build a series of compositional tasks tailored to the RL agent's current skill level. This automatic curriculum not only enables the agent to learn more complex tasks than it could have otherwise, but also selects tasks where the agent's performance is weak, enhancing its robustness and ability to generalize zero-shot to unseen tasks at test-time. We analyze why current environment generation techniques are insufficient for the problem of generating compositional tasks, and propose a new algorithm that addresses these issues. Our results assess learning and generalization across multiple compositional tasks, including the real-world problem of learning to navigate and interact with web pages. We learn to generate environments composed of multiple pages or rooms, and train RL agents capable of completing wide-range of complex tasks in those environments. We contribute two new benchmark frameworks for generating compositional tasks, compositional MiniGrid and gMiniWoB for web navigation. CoDE yields 4x higher success rate than the strongest baseline, and demonstrates strong performance of real websites learned on 3500 primitive tasks.

## 1 Introduction

Consider purchasing an airline ticket, logging in to a website, or buying movie tickets. These tasks can be completed by mastering a small set of basic manipulation skills (*primitives*), such as entering an appropriate text in a fill-in field, or selecting a date, and combining them in different ways to form complex, *compositional tasks* [42]. Humans can easily generalize between compositional tasks – purchase a ticket on an airline or fill out a form that they have not seen before, even when the task carries over several pages – but training autonomous agents to do this is far from straightforward. Yet, unlocking generalization across related tasks would pave the way towards autonomous agents that can automatically handle the details of completing wide variety of real-world user requests such as, "Buy me a plane ticket to Los Angeles leaving on Friday". The complexity and diversity of real environments make this a formidable challenge, especially due to the exponentially exploding action space and sparse rewards.

Generalizing across compositional tasks is challenging for several reasons, including the *tractability of training* [42]. Many compositional tasks require planning over an excessively long horizon while providing only sparse rewards, making it difficult for RL agents to learn. Presenting easy tasks

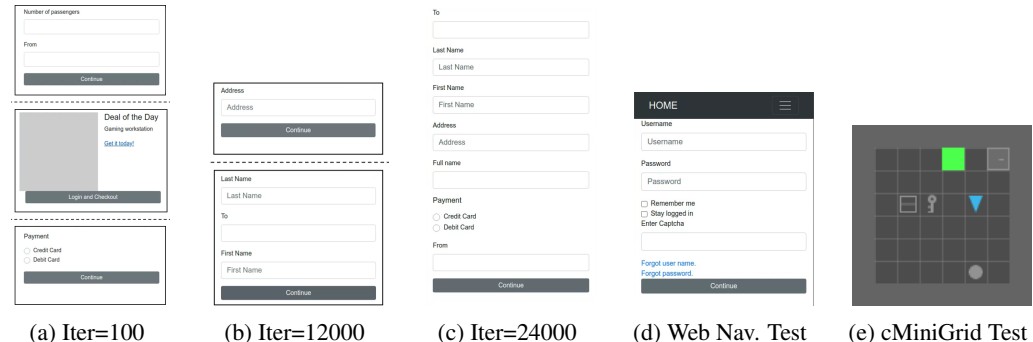

| (a) Iter=100 | (b) Iter=12000 | (c) Iter=24000 | (d) Web Nav. Test | (e) cMiniGrid Test |

Figure 1: Environments examples. Generated Web pages become more complicated over the course of training (a-c), unseen test "Login" website (d) and "Task List" minigrid (e) environments. More in Appendix A.13.

until the agent learns can be a solution, but it is not clear how to construct such a curriculum for complex compositional tasks [20]. Manually designing a pre-defined curriculum is tedious, and intractable. Domain randomization (DR) [41; 17] does not tailor the difficulty of the task to the ability of the agent. Automatically generating tasks tailored to the agent's learning progress is a promising approach, but as we will show, current approaches (e.g. [8]) suffer from fundamental limitations in the context of generating compositional tasks.

We present a new automatic curriculum generation algorithm, Compositional Design of Environments (CoDE), which jointly trains a Generator and a population of Learner agents. The generator is trained with a novel multi-objective reward function. Its first objective is to maximize the regret between the agents in the population and the best-performing agent, stabilizing regret estimation and making it less vulnerable to becoming stuck in local minima. Second, the generator is trained to adjust the task difficulty to match the learners' proficiency using an explicit difficulty incentive designed for use with compositional tasks. In this way, the generator builds more challenging environments when the learners are performing well, and reduces the difficulty when the learners are struggling. We demonstrate that this difficulty incentive addresses degenerative edge cases present in prior work.

By automatically searching for tasks on which the learners are performing poorly, the generator makes the learners more robust, enabling them to generalize to unseen tasks at test time. This ability is necessary for many real-world tasks, such as web navigation. Prior work on web navigation relied on collecting demonstration data from a series of training websites [36; 21], or training a separate policy for every single website [17]. These techniques cannot scale to a production-quality system, because they would require re-training the agents every time an airline company updated its site. In contrast, we show that CoDE produces robust agents with general form-filling and navigation skills that can be applied zero-shot to novel tasks at test time.

Another challenge in learning compositional tasks is that the *existing benchmarks do not match real-life complexities*. For example, real websites are orders of magnitude more complex than existing benchmarks [36; 21]. To address this problem, we introduce two new benchmark tasks for compositional task generation, which we are releasing in open-source. The first provides a way to automatically generate Minigrid [6] navigation tasks which include subtasks such as locating a key to open a door. The second, generative MiniWoB (*gMiniWob*), focuses on web navigation and manipulation (form-filling) tasks and enables a generator to construct increasingly complex form-filling websites, spanning multiple pages, out of common design primitives such as *navigation bars*, *product carousels*, *item decks*, and *item carts* (Figure 1). The evaluation environments in gMiniWoB are orders of magnitude more complex than MiniWoB, the prior web navigation benchmark [36].

This paper makes the following contributions. First, we formally introduce compositional tasks by drawing a connection to the Petri Nets graph formalism, and showing its relationship to POMDPs. We then analyze why prior techniques for automatically generating a curriculum of tasks are insufficient for compositional tasks, and propose a new algorithm, CoDE that addresses these weaknesses. We build two new automatic task generation environments spanning simple navigation tasks and web navigation, and release both in open-source. We demonstrate strong empirical results across both domains. In the context of web navigation, we show that CoDE generates a curriculum of increasingly challenging websites. Resulting agents successfully generalize to complex, unseen sites at test time, and without additional training solve wide range of form-filling tasks from flight purchases to login in. CoDE agents solve the most difficult tasks with $\approx 90\%$ success rate, 4x improvement

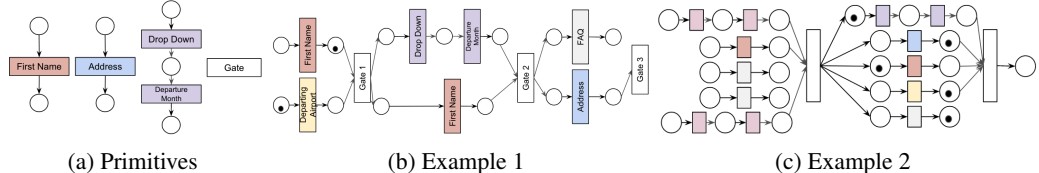

|  (a) Primitives  |  (b) Example 1  |  (c) Example 2  |

Figure 2: Examples of compositional task primitives (a) and two different compositional tasks composed using the same primitives (b-c). Note, passive primitives (in grey) that do not lead to the task progression and serve as a distraction for the agent. Difficulty budget for the generator is based on the number of primitives used.

over the strongest baseline. Lastly, we demonstrate that the method scales up to real websites. The implementation of CoDE and gMiniWoB framework are available in open source at `https://github.com/google-research/google-research/tree/master/compositional_rl`.

## 2 Related work

**Compositional task representation:** Compositional tasks [3] have been represented as a vector sum of primitive skills [9], or sequence of subtasks [30]. The dependency graph representation [37; 38] considers a task as a graph of subtasks equipped with dependencies as logical expressions. We relax the definition of the compositional tasks to represent real-world challenges. First, we allow primitives that do not lead to task completion. Second, we allow for partially observed environments. Third, we formalize the task distribution with a family of Petri Nets [34], which have also been used to describe robot tasks [10], planning tasks [27], and workflows [1]. We use PNs with color and hierarchy, structured similarly to the workflows, but allow dangling nodes. Beccuti et al. [4] map PNs to MDPs, while we propose mapping between a set of PNs to a set of POMDPs.

**Curriculum Generation:** A variety of curriculum learning methods exist which do not generate whole environments. For example, Florensa et al. [13] train a Generative Adversarial Network (GAN) to generate a curriculum of goal images for a goal-conditioned policy to reach. Other methods focus on choosing which task out of a set of pre-defined tasks to present next [29; 45; 12]. Another line of research studies creating curricula where a target task is already given [40; 31]. Multi-agent training can also be effective for automatically generating a curriculum [19; 22; 16; 32; 39; 28]. Czarnecki et al. [7] takes an alternative approach and generates a curriculum over agents instead of tasks. Finally, Active Domain Randomization [25; 33] learns to set the parameters of an environment simulator in order to provide a curriculum of increasingly difficult environments. In contrast, we focus on sequentially constructing an entire environment or task, composed of multiple subtasks, without a target task is explicitly given.

**Environment Generation:** POET [43; 44] generates terrains for a 2D walker with a population of environment-generating adversaries. Campero et al. [5] train a teacher to propose navigation tasks based on a length threshold. Most closely related to our work is PAIRED [8], which generates environments that induce maximal regret between a pair of agents. However, PAIRED only demonstrates results on simple gridworlds, and does not expand to complex, compositional tasks. We analyze several deficiencies in PAIRED that inhibit its use for the compositional setting, including a degenerate case where it fails to work at all. Our proposed algorithm addresses these deficiencies. To the best of our knowledge, we are the first work to attempt to automatically generate compositional tasks, and the first to apply environment generation to the problem of web navigation.

**Web navigation benchmarks and tasks:** MiniWoB [36] and MiniWoB++ [21] environments contribute a fixed set of manually curated toy websites, and train agents using expert demonstrations for each website. This cannot scale effectively to cover the large variety of real-world websites, and cannot adapt to changing websites. Further, these methods failed to solve complex tasks such as flight booking or social media interaction. Using a manually scheduled curriculum [17] improves upon these results, but does not adapt to the agent's progress or generalize to unseen sites. This work differs in several ways. First, we introduce a new framework, gMiniWoB, that allows generating complex websites on-the-fly with tunable difficulty levels. Additionally, we do not rely on any expert demonstrations to augment sparse rewards. Most importantly, our web navigation agents generalize to unseen environments, an important requirement for real-world web navigation agents.

## 3 Compositional tasks problem definition

Compositional tasks are composed of subtasks, which in turn are composed of primitive skills. This type of problem can be formalized using a task dependency graph, in which dependencies between the

subtasks are represented as edges (see Figure 2). We model tasks in which the agent must complete several subtasks in order to trigger a phase transition in the environment, such as locating a key and unlocking a door in order to enter a new room, or completely navigating and correctly manipulating one web page in order to transition to the next one.

We assume that the navigation agent uses standard POMDP setup – a tuple $(\mathcal{S}, \mathcal{A}, \mathcal{T}, \mathcal{O}, r, \gamma)$, where $s \in \mathcal{S}$ are states, $o \sim \mathcal{O}(s)$ are partial observations, $a \in \mathcal{A}$ are actions, and $\gamma \in [0, 1)$ is a discount factor. At each timestep $t$, the agent selects action $a_t$ according to its policy $\pi$, receives reward $r(a_t, s_t)$, and the environment transitions according to $\mathcal{T}(s_{t+1}|s_t, a_t)$. Agents seek to maximize their *return*, the discounted sum of rewards: $R(\pi) = \sum_{t=0}^{T} \gamma^t r_t(a_t, s_t)$.

We use Petri Nets (PNs) [14; 11] framework to formally define *learnable compositional tasks*. PNs are directional graphs consisting *place* and *transition* nodes, and easily model different dependencies such as sequential, concurrent, choice, etc. *Places* are system states; *transitions* are points in the process that takes the system to the next state. *Edges* determine the dependencies between nodes. At execution time, PNs have tokens that propagate through the net between places and determine the system state. In PNs with colors (CPN) [14] tokens are assigned a value, and Hierarchical PNs [11] replace subnets with transitions.

We define a *primitive* as a workflow (a special type of PN) with multiple places/transitions where there is no parallelism (see Figure 2a). They can be combined in parallel or serial using another transition (ex: Figure 2b). The set of colors, $C$, represents data semantics that agent needs to place in the environment to complete the task (i.e. flight departure or return dates). In the form filling task, agent needs to complete all the relevant fields before proceeding to the submit button that presents the next page. To that end, we introduce a *gate*, a special transition state that completes a phase and moves the system to the next (e.g. move between the pages in web navigation, or move between rooms in navigation tasks). *Learnable compositional tasks*, is a family of a directed acyclic PNs with colors induced with a set of primitives $P_C$.

*Learnable tasks* described above map compositional tasks to POMDP $(\mathcal{S}, \mathcal{A}, \mathcal{T}, \mathcal{O})$ that define RL navigation agents, and generator architecture which allows the generator to generate tasks only with the structure defined by PN. See Appendix A.1 for more details on PetriNets and their connection to POMDPs.

## 4 Environment generation with minimax regret analysis

To train robust agents able to learn complex tasks, it would be helpful to generate a curriculum of tasks that are designed to be just outside of their current skill level. This requires training a Generator policy $\pi^G$ to construct an environment $\mathcal{E}$ that will challenge a learner policy $\pi^L$ (e.g. [43]). To find tasks that are solvable, but where the learner is weak, we can train the generator to maximize the *regret*. Regret is the difference between the return obtained by policy $\pi^L$ and the return that would have been obtained with the optimal policy $\pi^*$ in the same environment $\mathcal{E}$: $\text{REGRET} = R^{\mathcal{E}}(\pi^*) - R^{\mathcal{E}}(\pi^L)$. Because the optimal policy is not known *a priori*, PAIRED [8] approximates the regret by introducing a third *antagonist* agent, with policy $\pi^A$, and computing the regret with respect to it: $\mathcal{J}^{\text{PAIRED}} = R^{\mathcal{E}}(\pi^A) - R^{\mathcal{E}}(\pi^L)$. In order to generate a viable curriculum, regret minimization relies on the assumption that the reward function includes an incentive to complete the task more efficiently. In this case, the regret will be highest for easy tasks which could be completed in a few steps by the optimal policy, but which the learner policy $\pi^L$ fails to complete. Thus, regret is a useful objective for inducing a curriculum when it is possible for the generator to build impossible environments, and the reward includes incentives to complete the task in fewer timesteps.

**Degenerate case.** What happens if the assumptions listed above are violated? For example, consider the case where there is only a binary reward $r \in \{+1, -1\}$ depending on whether the agent correctly performs a subtask or not, and the generator is constrained such that it cannot construct impossible environments. These assumptions are both true in the case of the compositional task graphs we study in this paper. Consider the simplest possible task graph, the Chain MDP pictured in Figure 3. The agent only receives a positive reward if it reaches state $s_N$, and from state $s_i$

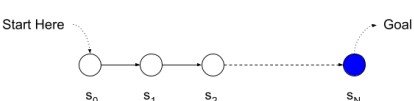

Figure 3: In chain environments, it becomes exponentially more difficult to reach the goal via random exploration as $N$ increases.

it can only reach state $s_{i-1}$ or $s_{i+1}$. The probability of reaching $s_N$ through random exploration is $P_{reach}(N) = O(p^N)$ (see Appendix Section A.2 for the proof), which decreases exponentially as more states are added. By our assumption, we know it is possible to reach $s_N$ and so the reward for the optimal policy is $R(\pi^*) = 1$. Then, maximizing REGRET $= 1 - R^{\mathcal{E}}(\pi^L)$ reduces to minimizing the probability that the learner obtains a reward, i.e. $P_{reach}(N)$. This can be accomplished by continuing to increase $N$. For compositional tasks, this means the generator will add as many elements to the environment as possible. In this case, maximizing regret will not lead to a curriculum, and could actually slow learning more than randomly generating environments with Domain Randomization.

**Sparse rewards.** Even if the assumption that $R^{\mathcal{E}}(\pi^*)$ is higher for easier environments is met, the regret signal used to train the generator can still be extremely sparse. Consider the case where both the learner and antagonist agent cannot solve the task. In this case, they never receive a positive reward, their rewards are equal, and the regret is zero. This gives the generator no signal with which to learn to decrease the difficulty of the environment to enable them to start learning. Once again, the generator is no better at generating a curriculum than randomly sampling environments.

**Lack of convergence and stalled training.** The proof that the PAIRED algorithm will produce a learner agent with minimal regret relies on the assumption that the game will reach a Nash equilibrium [8]. However, in practice gradient-based multi-agent RL has no convergence guarantees, is highly non-stationary, and will often fail to converge [23; 24]. If the game does not converge, then PAIRED minimizes regret with respect to the antagonist's policy, which only forces the learner to be as good as the antagonist. If the antagonist fails to improve, or reaches a local optimum, then the curriculum will stall and the learner cannot continue to improve.

## 5    Compositional Design of Environments (CoDE)

We present a new algorithm, Compositional Design of Environments (CoDE), for automatically generating a curriculum of compositional tasks. CoDE trains a generator to construct compositional environments out of primitives, and then uses those environments to train a population $\mathcal{P}$ of learner agents. The rewards obtained by the learners across environments are used to compute a multi-objective reward function to train the generator to produce an effective curriculum.

**Population-based regret estimation:** Despite issues with its estimation in prior work, regret is a useful incentive for environment generation because it encourages the generator to explore towards environments that promote variation in performance, indicating they have learning potential. Thus, the first component of our algorithm is a new, more flexible population-based estimate of regret (PopRegret) that is not subject to the problem of stalled training if the game converges to local optima. For each environment $\mathcal{E}$ constructed by the generator, each agent $p \in \mathcal{P}$ collects $M$ episodes in the environment. The regret is the difference between the average score of the population, and the score of the policy that achieved the highest average return over the $M$ episodes. Let $R_m^{\mathcal{E}}(\pi^p)$ be the return obtained by policy $p$ in episode $m$. Then:

$$\mathcal{J}^{\text{POPREGRET}} = \max_p \mathbb{E}_m[R_m^{\mathcal{E}}(\pi^p)] - \frac{1}{K}\sum_{i=1}^{|\mathcal{P}|} \mathbb{E}_m[R_m^{\mathcal{E}}(\pi^i)] \tag{1}$$

As long as any agent has higher performance than any other agent, the objective will continue to identify environments in which there is learning potential, preventing training from stalling. As the agents continue learning, the best-performing agent in the population more closely approximates the optimal policy, providing a stronger estimate of the regret. Further, the regret estimate is smoother and more consistently positive than the regret obtained via the maximum of a pre-selected agent, providing a more stable signal for the generator to learn to optimize.

**Difficulty budget:** Even with a more stable estimate of the regret, training can still be slow and unstable if the rewards are sparse, and potentially subject to the degenerate case identified in Section 4. To address these issues, we add an incentive for the generator to tune the difficulty of the environment to the current skill level of the agents. Let $N$ be the number of primitives added to the task graph, and $p = \arg\max_{p \in \mathcal{P}} R^{\mathcal{E}}(\pi^p)$ be the agent with the highest score in environment $\mathcal{E}$. Then:

$$\mathcal{J}^{\text{DIFFICULTY}}(\mathcal{E}) = (\mathbb{1}[R^{\mathcal{E}}(\pi^p) > \beta] - \mathbb{1}[R^{\mathcal{E}}(\pi^p) < \delta])N/N_{max} \tag{2}$$

where $\mathbb{1}$ is the indicator function and $\delta < \beta$ are reward thresholds for failure and success, respectively. Thus, if the best-performing agent gets a high score on the task, the generator gets a reward based on

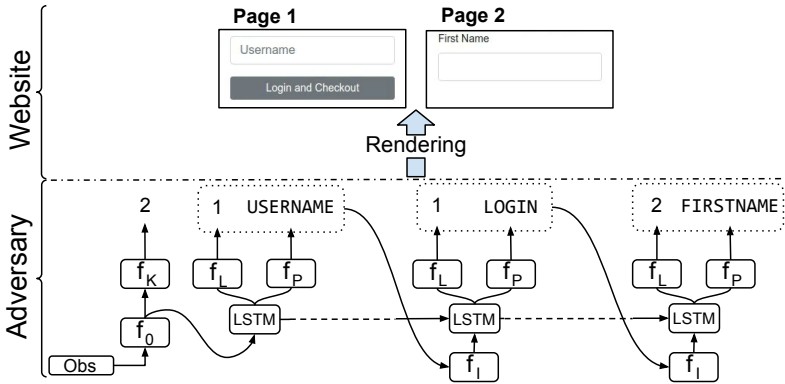

Figure 4: A sample rollout of the generator for a compositional web navigation task. An initial observation (Obs) is given at the beginning of the rollout. $f_0$, $f_K$, $f_P$, and $f_I$ networks encode initial observation, creating subtasks, primitives, and encoding LSTM inputs. In practice, we also use an additional independent network $f_L$ to create page indices where each primitive is assigned.

the number of primitives it added to the environment, $\frac{N}{N_{max}}$. However, if all the agents are performing poorly, the generator is penalized according to the number of primitives it added and receives $-\frac{N}{N_{max}}$. This makes adding more primitives risky, unless the generator can be sure that at least one agent in the population can solve the environment. It is thus incentivized to create tasks that challenge all but the best-performing agents in the population, while still decreasing the difficulty when the agents are not learning.

**Generator model:** The generator is a decoder designed to construct an environment $\mathcal{E}$ by placing a set of primitives $a \in \mathcal{A}$ within a set of subtasks $[\mathcal{W}_1, \cdots, \mathcal{W}_K]$. We assume a fixed maximum number of subtasks $K$, and allow for a up to $N$ primitives to be added to the environment. To control for complexity, empty subtasks with no primitives are allowed. The generator's policy $\pi^G(\mathcal{E}|o^G)$ is conditioned on an initial observation $o^G \sim N(0, I)$ sampled from the standard normal distribution, which allows it to diversify its design distribution (similar to a GAN generator [15]). It then constructs the environment according to:

$$\pi^G(\mathcal{E}|o^G) = \pi_{\mathcal{E}_w}(k|K) \prod_{i=0}^{n} \pi(a_i|a_{0:i-1}, j) \tag{3}$$

where $a_i$ corresponds to a primitive. The generator first encodes $o^G$ with a feed forward network $h_0 = f_0(o^A)$. The encoding $h_0$ is passed to another network $f_K$ that outputs a Categorical distribution $Cat(0, K)$, from which the number of subtasks is sampled $k \sim Cat(0, K)$. Given $k$ and initial state $h_0$, the generator executes an autoregressive LSTM policy to sample primitives $a_i$, through a network $f_P$ (see Appendix A.14 for more details). Thus, the generator is a stochastic, open-loop policy. Note that there is a SKIP action that does not add a new primitive, enabling the generator to reduce complexity. The finished design $\mathcal{E}$ is sent to a renderer module to generate the environment. Figure 4 shows an example of how the generator constructs an environment which the renderer converts to a website. Note that the generator is domain independent, and creates a generic compositional task or PN; indeed, we use the same model for both experiment domains.

**Difficulty budget implementation:** Rather than directly counting the number of primitives for the difficulty, we replace $N$ in Eq. 2 with the amount of probability the generator places on the SKIP action throughout generating the environment: $\hat{N} := -\sum_{i=1}^{n} \log \pi_{\mathcal{E}}(a_i = \text{SKIP}|a_{0:i-1})$. This allows for a more direct control over generator policy to optimize difficulty. We also scale Eq. 2 by the reward received by the best-performing agent to provide further signal to the generator.

**Algorithm:** Algorithm 1 shows the CoDE training procedure. Both the generator and learner agents are trained using RL, specifically A2C [26] with entropy regularization. For every training step, the generator constructs an environment $\mathcal{E}$, and the agents $p \in \mathcal{P}$ collect $M$ trajectories within $\mathcal{E}$. The learner agents are trained using the standard task related reward. To train the generator, we use the following multi-objective loss function that encourages the adversary to control the complexity of the environment by presenting "just-the-right challenge" for the agents in $\mathcal{P}$, where $\alpha$ is a hyperparameter:

$$\mathcal{J}(\mathcal{E}, \mathcal{P}) = (1 - \alpha) * \mathcal{J}^{\text{POPREGRET}}(\mathcal{E}, \mathcal{P}) + \alpha * \mathcal{J}^{\text{DIFFICULTY}}(\mathcal{E}, \mathcal{P}) \tag{4}$$

**Algorithm 1:** CoDE: Joint training of the generator and learner agents.

---

**Initialize** policies for generator $\pi^G$, and agents $\pi^p, \forall p \in \mathcal{P}$

**for** *all training iterations* **do**

    $\mathcal{E}_w \sim \pi_G(\mathcal{E}|o^G), o^G \sim N(0, I)$ ;           ▷ *Generator constructs environment*

    **for** $p = 1, \cdots, |\mathcal{P}|$ **do**

        **for** $m = 1, \cdots, M$ **do**

            $R_m^{\mathcal{E}}(\pi^p) \longleftarrow$ Collect rewards for agent $p$ in environment $\mathcal{E}_w$;

        RL update for agent $p$ using collected experience ;     ▷ *Train learner*

    Compute $\mathcal{J}^{\text{REGRET}}$ using Eq. 1;

    Compute $\mathcal{J}^{\text{DIFFICULTY}}$ using Eq. 2 ;

    RL update for generator using Eq. 4 as the reward ;     ▷ *Train generator*

---

## 6 Benchmark environments

We contribute two open-source frameworks for compositional tasks construction out of a set of primitives: *Compositional MiniGrid (cMiniGrid)* and *Generative MiniWoB (gMiniWob)*.

**cMiniGrid** is a compositional extension of MiniGrid navigation environments [6], in which agents interact with various objects. cMiniGrid contains the following subtasks: *pick up the key*, *unlock the door*, *pick up the ball*, *open the box*, *drop the ball in the box*, *reach the goal*. The generator chooses subtasks to add, and the renderer places objects. The problems have sparse rewards which are only given for completing a subtask, which each require many steps.

**gMiniWob** is a generative extension of miniWob and miniWob++ web navigation benchmarks, extending them to multi-page navigation and richer websites. gMiniWob enables generation of web navigation tasks consisting of multiple web pages (subtasks), each composed of primitives (see Figure 2). The renderer handles linking the pages via output primitives (e.g. by adding a submit button). Each primitive is represented using a Document Object Model (DOM) tree, and rendered using HTML. gMiniWoB implements 40 common web primitives, such as navigation bars, carousels, item decks, web forms, item carts, dropdowns, etc. The order in which primitives are added defines how a webpage will be rendered for the agent, facilitating controllable generation with a rich design set. Every primitive changes the DOM structure when the agent interacts with it. 26 of the 40 primitives are active, while the rest are passive (i.e. they do not relate to the task, like an ad banner). When the generator adds an active primitive, the renderer adds the corresponding piece of data to the agent's instruction set, increasing the complexity of the task. For example, adding a 'First Name' text box also adds a new *"firstname"* field into the instruction. See Appendix A.11 for all the primitives, and Appendix A.12 for the test set. Note that with 40 possible primitives across 10 web pages, gMiniWob provides the ability to generate $\approx 10^{14}$ distinct web navigation tasks.

As in [17], the web-navigating RL agent receives a text instruction containing various fields of information, and must interact with DOM elements, fill in the necessary information, and navigate through the pages to complete the task. The agent observes the DOM tree of the current page. Its actions are a tuple (element, field) that denotes inputting the field into the element. For example, in a flight booking task, given the instruction `{"Departure Date": "Friday", Destination Airport: "Los Angeles (LAX)", "Address": ...}`, the agent must first pick the textbox labeled "dest", find the corresponding field in the instruction (Destination Airport) and type the value ("Los Angeles (LAX)"). The agent receives a small penalty each timestep to encourage efficient navigation. Correctly filling out information within the page results in a positive reward, normalized over the total number of fields in the instruction (e.g. if there are $F$ fields, the agent receives reward of $1/F$). Agents receive a reward of 1.0 for completing the task, and -1.0 otherwise.

## 7 Evaluations

**Implementation:** Following Gur et al. [17], the learner policy is an LSTM based DOM tree encoder and a feed forward network to encode instruction fields. The policy outputs a joint distribution over

| Env. | Task | DOM Size | Instr. Size | LfD DOM[21] | LfD Vis.[36] | CL [17] | ALP [32] | PAIRED [8] | CoDE |
|---|---|---|---|---|---|---|---|---|---|
| MiniWoB [36] | password | 11 | 1 | 0% | 100% | 100% | | 100% | |
| | enter-text | 6 | 1 | 0% | 100% | N/A | N/A | N/A | 100% |
| | dynamic | 6 | 1 | 0% | 100% | 100% | | 100% | |
| gMiniWoB | login | 35 | 5 | | | 3% | 23% | 20% | **92%** |
| | address | 38 | 7 | | | 0% | 15% | 8% | **98%** |
| | payment | 49 | 5 | N/A | N/A | 0% | 7% | 16% | **93%** |
| | flight | 60 | 7 | | | 0% | 2% | 20% | **95%** |
| | shopping | 183 | 12 | | | 0% | 6% | 6% | **95%** |

Table 1: CoDE performance on MiniWoB form-filling and gMiniWob test environments compared to several baselines. DOM Size and Instruction Size illustrate the complexity of the environment. Learning from Demonstration (LfD) and Curriculum Learning (CL) methods train a separate model for each test environments, and the results are reported from the previous publications. PAIRED, ALP and CoDE train a single model that is evaluated across all benchmark tasks. CoDE generalizes to all tasks, and outperforms all baselines.

elements and fields by measuring pairwise similarities between element encodings and instruction fields. We compute the state-value by using the marginal distribution of elements as attention weights over element encodings and passing the context vector through a feed-forward network. CoDE is implemented using ACME [18] with TensorFlow [2] open-source libraries. The training is done on a single CPU, requiring about a week of training. The results reported are averaged over 5 seeds.

**Baselines:** We compare to: a) *PAIRED* [8]: for a fair comparison, we limit the size of the CoDE population $|\mathcal{P}| = 2$, since PAIRED uses two agents to estimate regret; b) *ALP* [32]: we use absolute difference between rewards of a single navigator at timestep $t$ and $t-1$ as the reward for the adversary without any difficulty bugdet; c) *Domain Randomization (DR)*: [35; 41] we sample the number of subtasks $k$ and the primitives from uniform distributions. d) *Curriculum Learning (CL)*: based on the state-of-the-art web navigation RL agent Gur et al. [17]. We adapt this method to zero-shot environment generation where we are not given a specific website but a set of design primitives. We randomly sample each primitive w.r.t. a probability $p$ where $p$ is initialized with a small number and scheduled to reach 1.0 during training; d) Learning from Demonstrations for web navigation from DOM inputs (LfD DOM) [36] and image (LfD Visual) [21].

**Web navigation evaluation:** We evaluate our models on MiniWoB [36], and a suite of test environments implemented in gMiniWoB ('Login', 'Enter Address', 'Flight Booking', 'Enter Payment', and 'Shopping' websites). Each environment comes with 4 different difficulty levels by gradually adding more primitives. These environments are never explicitly presented to agents during training, so performance measures how well agents can generalize to unseen websites at test time. To train CoDE we use 40 hand-engineered primitives, included with gMiniWob which are also used as building blocks for the test environments. Finally, we conduct an experiment based on real websites, where the generator chooses from among 3500 primitives scraped from the web. The primitives were extracted from 200 different websites in the password change, food ordering, and shopping domains. We evaluate agents' performance on a held out set of real websites, never presented during training.

## 7.1 Results

**Overall performance:** CoDE outperforms the baselines across the benchmarks, both in web navigation and cMiniGrid. CoDE is at least 4x more successful at completing web navigation tasks than the strongest baseline (see Table 1), and reaches more than 90% task success across all difficulty levels (Figure 7). On cMiniGrid environments, CoDE has nearly 3x the success rate of PAIRED, solving 45% vs 16% of tasks (Figure 5), demonstrating CoDE's versatility across very different domains. We present additional results and ablation study in Appendix A.3.

**Regret minimization with sparse reward:** In cMiniGrid rewards are sparse, and frequently no agents in the population receive reward, resulting in zero regret and absence of the training signal. As predicted, under these conditions maximizing the regret leads to poor performance (Figure 5). Training stalls, and the agents do not achieve high reward. In contrast, adding the difficulty objective significantly improves results.

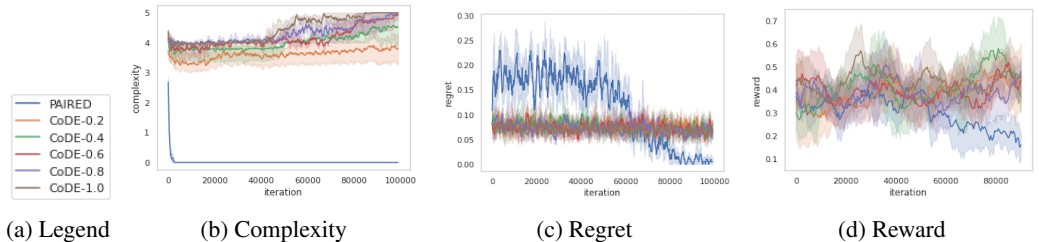

(a) Legend      (b) Complexity      (c) Regret      (d) Reward

Figure 5: cMiniGrid experiments. Due to sparse rewards, the PAIRED algorithm fails to generate complex environments or train agents with high reward. Adding difficulty incentive (with increasing $\alpha$) enables both.

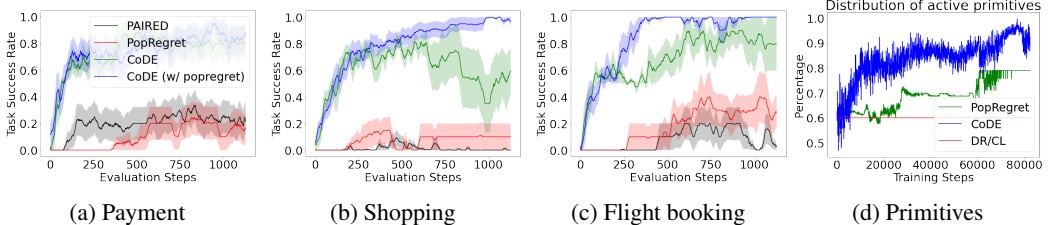

(a) Payment      (b) Shopping      (c) Flight booking      (d) Primitives

Figure 6: Comparison of PAIRED [8] and CoDE averaged over 4 difficulty levels. (f): Percentage of active primitives over training steps (see Appendix A.6 for more details).

**Importance of difficulty budget**: Difficulty incentive leads to a large improvement in performance of CoDE over the baselines on the web navigation tasks (Figures 6 and 7). When agents are struggling to learn and have very similar rewards, the regret becomes very small. This uninformative signal makes it difficult for the regret-based methods (PAIRED and PopRegret) to learn. On the other hand, CoDE provides a clear signal to the generator that improves performance significantly over all tasks under consideration. Detailed results are in Appendix A.8, and further ablation studies in Appendix A.10 investigate the effect of the budget weight $\alpha$.

**Importance of better regret estimation**: While the difficulty incentive provides a large boost to performance, it has a limitation in that it may not always be straightforward to calculate the difficulty of a generated task (for example, if the number of primitives added does not correspond directly to difficulty). In this case, it is necessary to rely on more general objectives such as regret, and stablizing the regret objective becomes important. We demonstrate, in two ways, that the proposed population-based regret estimate in Eq. 1 (PopRegret) provides an advantage over the regret estimation in PAIRED (Figure 6). First, while PAIRED training stalls, especially for difficult tasks such as shopping and flight booking, PopRegret agents learn more effectively. Second, we assess the contribution of PopRegret to the CoDE algorithm by comparing CoDE with no PopRegret (i.e. using only the difficulty incentive) to the full algorithm. While relying solely on the difficulty incentive is sufficient for easier tasks like payment, it does not perform as well as CoDE in more complicated tasks.

**Curriculum generation:** CoDE produces a curriculum of emerging complexity, estimated by the number of active and passive primitives. CoDE starts around 60% random selection of primitives, and gradually generates more primitives while improving agent's performance (Figure 6d). Even as the complexity of the environments continues to increase, CoDE agents still perform consistently well without degrading performance (Figure 7). CL's performance drops significantly due to ignoring agents' skill level and making environments that are too challenging for agents to complete, therefore ignoring their ability to learn. DR performs poorly because the randomly generated environments are so complex that the agents never even begin learning.

Next, both number of active and passive primitives increases (Appendix A.6), and the distribution of the primitives shifts to more complex and relevant primitives later on (Appendix A.5). Learning a web page with more passive primitives is a relatively easier task than a page with more active primitives, because passive primitives either add noise and should be ignored by the agents, or are used by agents only to navigate to another page.

Active primitives not only increase DOM tree size, but the number of instruction fields as well, making the matching between elements and instruction more challenging.

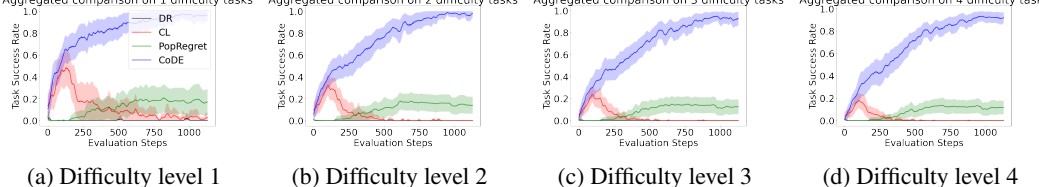

| (a) Difficulty level 1 | (b) Difficulty level 2 | (c) Difficulty level 3 | (d) Difficulty level 4 |

Figure 7: Aggregated task success rate comparison of CoDE and baseline models on test environments with increasing difficulty levels. See Appendix A.8 for detailed results.

CoDE agents continue to improve performance thanks to the ability to tune the difficulty of the environments to agents' proficiency.

**Real website evaluation:** CoDE successfully scales to real-world web navigation with generator choosing from a almost 100 times larger set of primitives, yielding $\approx 10^{31}$ possible environments. The trained agents generalize to unseen environments (Table 2) constructed out of real web data. As the distribution of real primitives mainly focus on username, password, and address fields, we observe a high evaluation success on the Login and Address tasks. Even though we used the same primitives for the Flight as above results, we observe a significant drop in performance which can be attributed to skewness of the primitive distribution.

|  | 1 | 2 | 3 | 4 |
|---|---|---|---|---|
| Login | 62% | 63% | 55% | 44% |
| Address | 36% | 25% | 29% | 38% |
| Payment | 23% | 18% | 12% | 9% |
| Shopping | 44% | 36% | 34% | 31% |
| Flight | 29% | 25% | 31% | 23% |

Table 2: CoDE performance on real websites across tasks and difficulty levels.

## 8 Broader impact, limitations, and future work

PN formalism (Appendix A.1) is useful beyond this work, as it exposes compositional task topology. We use expanded workflow model, which although expressive for a rich set of tasks seeded with only a handful of primitives, is still only one topological class of tasks. Further investigation is needed into the correlation between compositional task toplogies and methods. Do these methods work for tasks with alternate routes? Our preliminary investigation shows that the difficulty budget is not appropriate for tasks without a goal state (terminal state in PNs), regardless if the goal is not-reachable, or the task is open-ended. What methods can enable learning compositional tasks with difficult topologies?

Next, we evaluated the model trained with real website primitives on test set with hand-engineered primitives, with surprising results: success rate ranging from 90% on address tasks to 0% in a simple MiniWob enter-text. While expecting generalization across primitives is a tall ask, the results show that in some cases it is attainable. What are the circumstances when the primitives generalization is possible, and how can we develop models that expand the repertoire of primitives?

## 9 Conclusion

We present, Compositional Design of Environments (CoDE), a novel algorithm for zero-shot generalization compositional task learning. First, the Petri Nets formalism enables definition of the learnable task built from the few selected primitives, algorithmic creation of RL agents and generator and provides tools for toplogocial analysis of the learnable tasks. Second, we introduced two new objectives for curriculum generation: an improved, population-based estimate of regret, and a difficulty objective tailored to the problem of generating compositional tasks. Next, a proposed domain-agnostic generator architecture builds new hierarchical tasks out of primitives. To facilitate research on compositional tasks and environment generation, we controbuted two open-source benchmarks. Finally, we demonstrated that CoDE generates a curriculum of emerging complexity, successfully trains agents outperforming baselines 4x, and results in a *single* RL agent capable of filling out generic forms and completing variety of tasks on real websites.

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
