# A  Appendix

## A.1  Petri Nets formalism

Petri Nets (PNs), directional graphs consisting *place* and *transition* nodes, model concurrent systems with synchronous and asynchronous processes. *Places* are system states; *transitions* are points in the process where a user or an agent interacts with the system to perform an action and transition the system to the next state. *Edges* determine the dependencies between nodes. At execution time, PNs have tokens that propagate through the net between places. When a token reaches a place adjacent to a transition, if the transition is enabled and fires, the token is consumed, and new one is placed at the successor place. Token placement determines the system state. PNs easily model different dependencies: sequential, concurrent, choice, fork, join, mutual exclusion etc. In PNs with colors (CPN) [14] tokens are assigned a value, and Hierarchical PNs [11] replace subnets with transitions.

We focus on tasks that are compositions of subtasks, sets of primitives that need to be completed in order to proceed to the next stage. A *primitive* is a (simple or serial) workflow-net with colors and multiple places/transitions where there is no parallelism (see Figure 2a). They can be combined in parallel or serial using another transition. The set of colors, $C$, represents data semantics that agent needs to place in the environment to complete the task (i.e. flight departure or return dates). We assume that for each color in $C$ there is at least one primitive associated with it. Let $P_C$ be set of all primitives wrt color set $C$. For example, primitives are widgets that correspond to different ways of inputting departure date (as a fill-in field, or a selection from a calendar). Primitives contain an initial place (a state before the agent manipulates the environment), manipulation sequence, and end place (a state of the environment post interaction). In the instance of departure field primitive, an initial place is an empty date field, and the end place is date field with a departure field from the profile (Figure 2a). In the form filling task, agent needs to complete all the relevant fields before proceeding to the submit button that presents the next page. To that end, we introduce a *gate*, a special transition state that completes a phase and moves the system to the next (e.g. move between the pages in web navigation, or move between rooms in navigation tasks). We also partition the primitives in *active*, which make progress towards task completion, and *inactive* that are distractions and do not contribute towards the progress. *Learnable compositional tasks*, used here, is a family of a directed acyclic hyper PNs with colors induced with a set of primitives $P_C$ and the following properties. PN has at least one gate. All primitives sampled from $P_C$ must be reachable from a gate, but only active primitives must be predecessors to a gate. A *page* is a subnet consisting of all primitives leading out of a gate.

*Learnable tasks* described above map compositional tasks to POMDP $(O, A, T, R)$ that define RL navigation agents, and generator architecture which allows the generator to generate tasks only with the structure defined by PN.

POMDP hidden state is a PN configuration computed as location of all tokens. The POMDP transition function maps directly to place $\rightarrow$ transition $\rightarrow$ next place in PN. Observations consist of the set of colors, which provide the data for the task and are renderings of all primitives in a current page. Each transition node in PN task maps to a POMDP action. Available actions directly match to enabled transitions. Each action is represented as a tuple (element, field) that denotes acting on the element using the field as an input; i.e. typing the value of the field into the element. Finally, the rewards associated with the transitions as follows. At every time step a transition fires. Agent receives a small penalty each timestep to encourage efficient navigation. Emission of new tokens after a successful firing results in a positive potential-based reward for the agent, normalized over the total number of transitions. Agents receive a task success reward (1.0 or -1.0) when PN reaches the final state or times out. As an example, consider a flight booking task where the agent is given an instruction {"Departure Date": "Friday", Destination Airport: "Los Angeles (LAX)"}. The agent first picks a field (e.g. destination airport) and finds the corresponding text box in the page; then the corresponding value ("Los Angeles (LAX)") is typed into the text box. If this value is correct, the agent receives a positive reward of $1/2$ where 2 is the number of fields in the instruction. Learner RL agent is parameterized by a network $\pi(a_t|s_t; \Theta_i)$, that maps an observation $s_t$ to output action distribution to maximize the cumulative discounted reward, i.e., $O = \sum_{t=0}^{T} \gamma^t r_t$ where $r_t$ is the reward at time step $t$, $\gamma$ is a discount factor, and $T$ is the length of an episode.

To create generator the networks $f_P$ and $f_L$ build the task by sampling next primitive $a_i$ directly from the PN set of primitives, and its page locations $b_i$. Given the structure of the compositional task that

| | Login | Address | Payment | Shopping | Flight |
|---|---|---|---|---|---|
| CoDE | 92% | 98% | 93% | 95% | 95% |
| CoDE ($\beta = -0.2, \alpha = 0.8$) | 86% | 100% | 90% | 96% | 87% |
| CoDE ($\beta = 0.2, \alpha = 0.8$) | 57% | 24% | 22% | 33% | 10% |
| CoDE ($\beta = -0.2, \alpha = 0.9$) | 82% | 94% | 82% | 88% | 70% |
| CoDE ($\beta = 0.2, \alpha = 0.9$) | 50% | 28% | 43% | 25% | 15% |
| CoDE ($M = 4, \alpha = 0.8$) | 66% | 71% | 69% | 71% | 49% |
| CoDE ($M = 4, \alpha = 0.9$) | 84% | 96% | 94% | 91% | 85% |
| CoDE ($M = 6, \alpha = 0.8$) | 62% | 47% | 65% | 39% | 27% |
| CoDE ($M = 6, \alpha = 0.9$) | 60% | 68% | 54% | 69% | 34% |

Table 3: Ablation study of CoDE using different $\beta = \delta$ and $\alpha$ hyper-parameters. Using positive $\beta = \delta$ gives more conservative designs, CoDE is robust to different $\alpha$ values, and using larger number of episodes $M$ gives worse performance.

we used here, location and identity of each primitive is determined with its id, page location, and time step. Compositional tasks with different topologies will need more sub-networks to determine the location of the selected primitive.

PNs define the structure of learnable tasks induced with primitives, and map directly to POMDPs. The PN formalism allows us to reason about and sample related tasks in a principled way. POMDPs define a learning problem for the RL agent, which acts on the PNs to complete the task. Our goal is to train a single RL agent that solves the set of POMDPs. In the rest of the paper, generator designs PN compositional task and the corresponding POMDP, and the learner agent learns to solve it.

### A.2 Probability of successfully reaching the goal in a Chain MDP

Consider the following chain MDP (Figure 3) where the agent starts at the leftmost state ($s_0$), can move right or left, the goal ($g = s_N$) is the rightmost state, and the reward is {+1, -1} depending on if the goal is reached within $N + 2L$ steps. Let's assume that initially, $p$ is the probability of taking a right action. Reaching the goal at state $N$ via random exploration is

$$P_{reach}(N) = \sum_{t=0}^{L} P(\text{N+t right action and t left action}) \tag{5}$$

$$= \sum_{t=0}^{L} C(N + 2t, t) p^{N+t} (1 - p)^t \tag{6}$$

$$= p^N \sum_{t=0}^{L} C(N + 2t, t) (p(1 - p))^t \tag{7}$$

$$\leq p^N (1 + p - p^2)^L \tag{8}$$

where $L \geq N$ and the last line comes from $(1 + x)^n = \sum C(n, k) x^k$, $C(n + 2k, k) \geq C(n, k)$ for every $n > 0$. In the simplest case where $L = 0$, this becomes $p^N$.

### A.3 Additional Experiments

In Table 3, we present ablation studies for various hyper-parameters in CoDE. We experiment with positive and negative reward thresholds $\beta = \delta$ and different numbers of rollouts $M$ per training iteration. For each ablation, we also sample $\alpha$ from $\{0.8, 0.9\}$.

$\beta = \delta$ is an important hyper-parameter that needs careful tuning for the best performance but CoDE still outperforms baselines for different values. We observe that using a relatively high reward threshold (0.2) causes the adversary to become more conservative and increase complexity only when navigators are performing very strongly. Using a larger number of episodes can decrease performance.

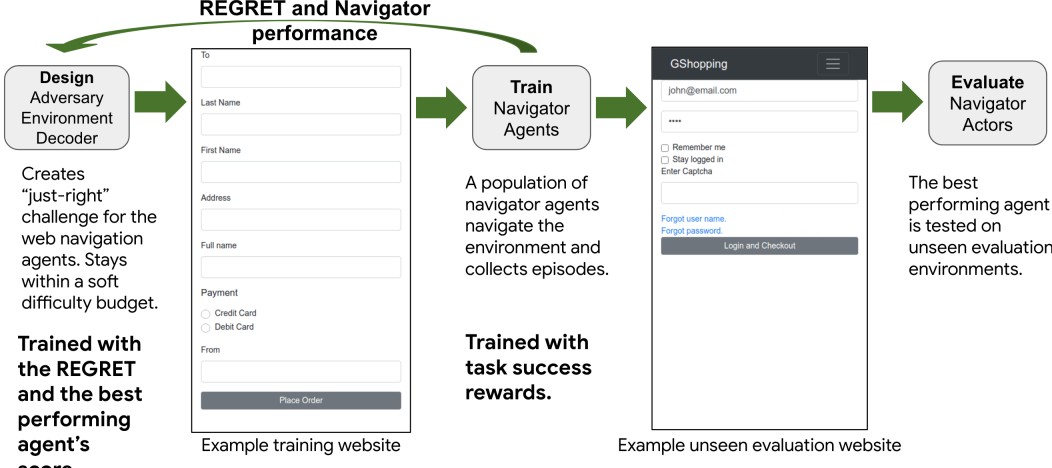

Figure 8: Training workflow. The adversary unrolls an environment, adding one element at the time of each page. That environment is passed on to a population of navigation agents under training. The navigators collect several episodes in the environment and their returns. The weight of the navigator agents are updated w.r.t. their returns. And the adversary weights are updated w.r.t. regret estimate and budget estimate using the best performing agent.

## A.4 Training flow

In Figure 8, we illustrate the high level workflow of the CoDE.

## A.5 Distribution of Primitives During Training

During training, the distribution of primitives become more skewed towards active primitives early on (as shown in Figure 6d), but as the environments get more challenging, more passive primitives are introduced as well (Figure 9). What we observe from the histograms in Figure 9 is that new primitives are slowly introduced while the ranking of the primitives is also slightly changed.

## A.6 Active and Passive Primitive Frequencies

In Figure 10, we present frequencies of active and passive primitives during training. With CoDE number of both active and passive primitives increase resulting in more complex websites.

## A.7 Creating primitives from DOM templates

In Algorithm 2, we outline the process for generating a new primitive from a given DOM HTML template. A DOM template is a piece of self-contained HTML with variables. We generate new primitives by assigning values to variables in DOM templates.

## A.8 Detailed Results on Test Environments

We detail the aggregated results in Figure 7 and present performance of agents across tasks and difficulty levels (Figure 11). On the easiest level of tasks, CL achieves slightly lower performance than CoDE early in the training while as the task difficulty increases, the gap becomes more apparent.

## A.9 cMiniGrid Details

In cMiniGrid, we use 5 different subtask primitives (Figure 12): (i) Pickup the key, (ii) Open the door, (iii) Pickup the ball, (iv) Open the box, and (v) Drop the ball. The adversary designs a grid by selecting

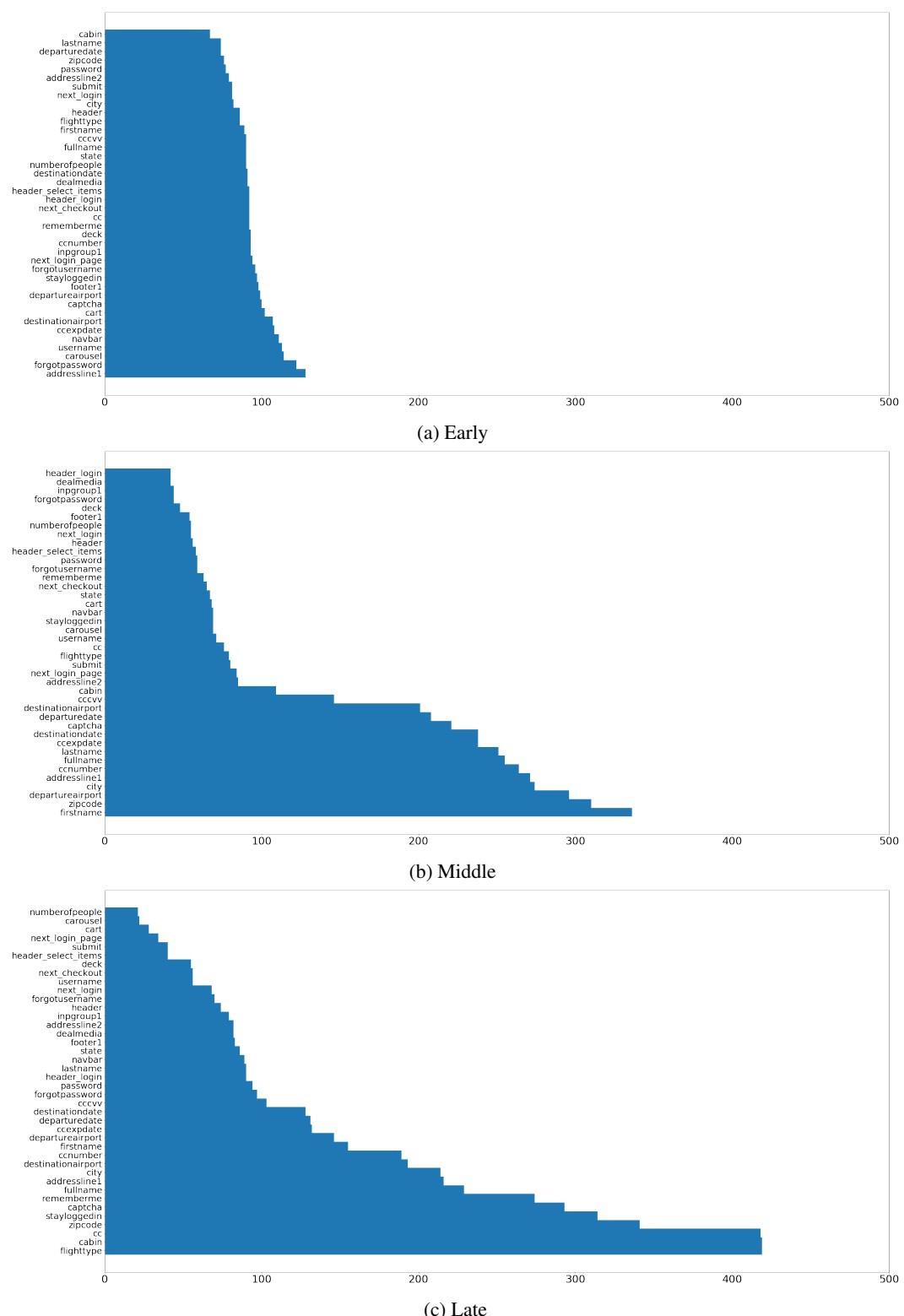

Figure 9: Histograms of primitives from early, middle, and late snapshots of the training.

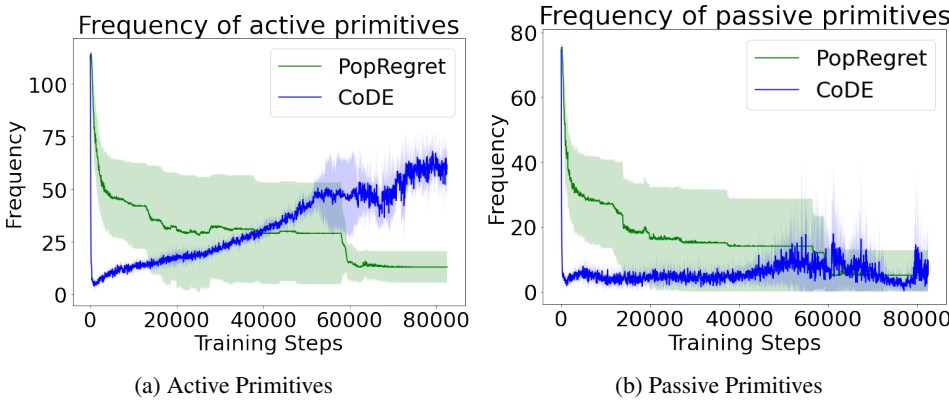

(a) Active Primitives  (b) Passive Primitives

Figure 10: Frequencies of active and passive primitives during training.

---

**Algorithm 2:** Generating a new primitive from a DOM HTML template.

---

**Input:**$D = (D_n, D_e)$: A DOM template, an HTML sub-tree with elements $D_n$ and edges $D_e$
**Input:**$V \subset D_n$: A list of elements that correspond to variables in $D_n$
**Input:**$A_{v,i}$: A list of variables $A_{v,i}$ for an element $v \in D_n$
**for** $v \in V$ ▷ *Iterate over elements.*
**do**
   | Flip a coin. If it is heads, $D_n \longleftarrow D_n \setminus \{v\}$. ▷ *Add/remove an element.*
**for** $v \in D_n$ ▷ *Iterate over elements.*
**do**
   **for** $a \in A_{v,i}$ ▷ *Iterate over variables for element v.*
   **do**
     | Flip a coin. If it is heads, sample and assign a value for $a$. ▷ *Add/remove an variable.*
   If there is at least one variable remaining for element $v$, $D_n \longleftarrow D_n \setminus \{v\}$.

---

a set of subtasks and cMiniGrid stitches them together according to the global subtask workflow. In Figure 12, we present the global subtask workflow and a sample design. In this example, the adversary selects *pickup key, pickup ball*, and *open box* subtasks. The sample workflow is generated from the set of subtasks while respecting the global workflow. Finally, cMiniGrid randomly places corresponding objects and the agent to empty cells in the grid. We assume that there is only a single room in the grid and the goal is always included in the set of subtasks. We use the global subtask workflow as our testbed to evaluate the performance of the population of agents.

Different from designing web pages, the order of subtasks has no effect in cMiniGrid. The adversary network is similar to the one used in gMiniWoB except that there are no predictions for number of cells and cell locations as objects are randomly assigned to cells. To train the population of agents, we use a 3-layer Convolutional Neural Network (CNN) for image and a 2-layer Multi Layer Perceptron (MLP) for position observations, respectively. A final 2-layer MLP is applied to image and position encodings to generate a distribution over actions. We use ReLU activations in both networks.

### A.10 Comparison of $\alpha$ in budget weighting

In Figure 13, we plot results where CoDE is trained with different $\alpha$ weights. We observe that as the $\alpha$ increases, we get consistent improvements illustrating the importance of the novel budget loss. With very small $\alpha$, the performance also degrades over time as the adversary is not able to adapt to skill level of agents.

### A.11 Web Environment Design Primitives

In Table 4, we present the list of design primitives, corresponding templates, types, and descriptions.

**Design Primitives and Their Descriptions**

| Design Primitive | Design Template | Active/Passive | Description |
|---|---|---|---|
| addressline1 | input | active | Main address information |
| addressline2 | input | active | Secondary address information |
| cabin | multi-selection | active | Multiple cabin options |
| captcha | input | active | Captcha information |
| carousel | carousel | passive | Items with images in a carousel with previous and next buttons |
| cart | cart | passive | Items in a product cart with promo code information |
| cc | multi-selection | active | Multiple credit card type options |
| cccvv | input | active | Credit card CVV information |
| ccexpdate | input | active | Credit card expiration date information |
| ccnumber | input | active | Credit card number information |
| city | input | active | City address information |
| dealmedia | media | passive | Product media with image, label, and link |
| deck | deck | passive | Multiple product decks with image, label, and link |
| departureairport | input | active | Departure airport information |
| departuredate | input | active | Departure date information |
| destinationairport | input | active | Destination airport information |
| destinationdate | input | active | Destination date information |
| firstname | input | active | First name information |
| flighttype | multi-selection | active | Multiple flight type options |
| footer | footer | passive | Footer with links and information |
| forgotpassword | link | passive | Link with forgot password context |
| forgotusername | link | passive | Link with forgot username context |
| fullname | input | active | First and last name information |
| header | label | passive | Generic header |
| header_login | label | passive | Header for login form |
| header_select_items | label | passive | Header for item selection |
| inpgroup | input | passive | Generic input with default search context |
| lastname | input | active | Last name information |
| navbar | navigation bar | passive | A navigation bar with a menu |
| next_checkout | button | passive | Next button with checkout context |
| next_login | button | passive | Next button with login context |
| next_login_page | button | passive | Next button with login context |
| numberofpeople | multi-selection | active | Multiple number of people options |
| password | input | active | Password information |
| rememberme | selection | active | Checkbox with remember me context |
| state | input | active | State information |
| stayloggedin | selection | active | Checkbox with stay logged in context |
| submit | button | passive | Submit button |
| username | input | active | Username information |
| zipcode | input | active | Zipcode information |

Table 4

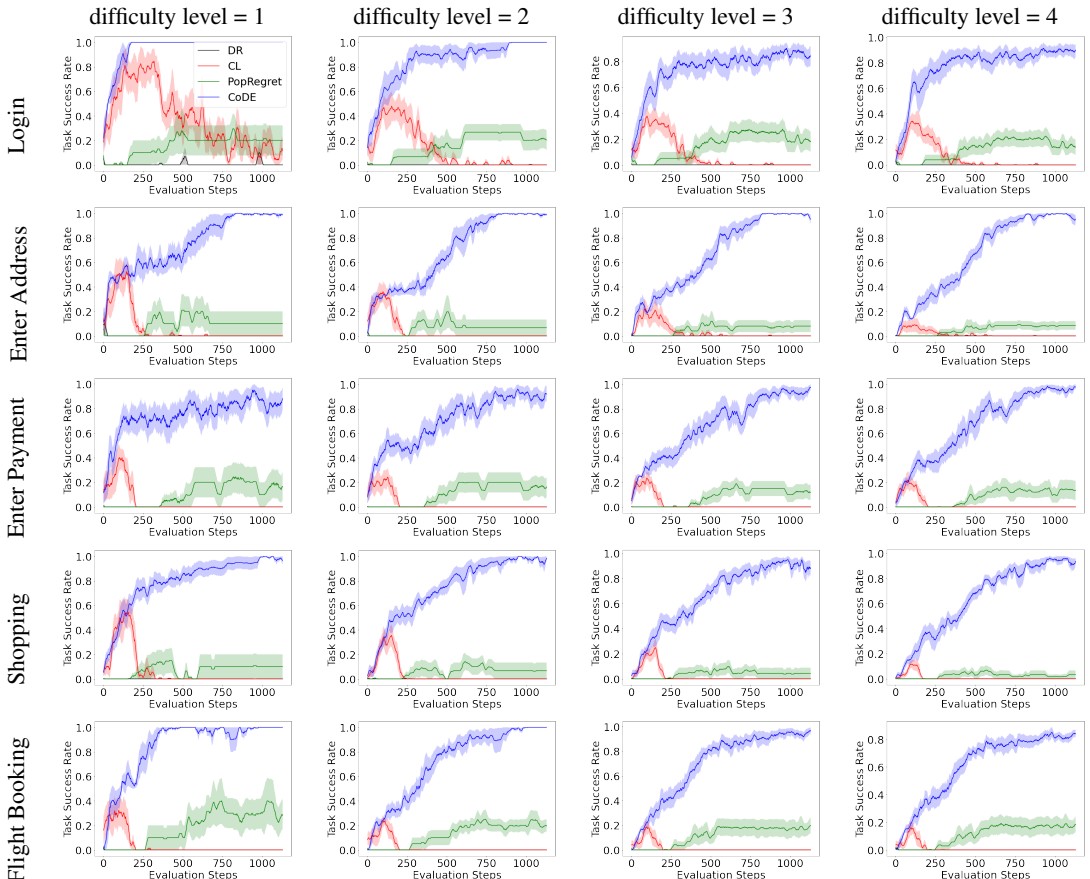

Figure 11: Task success rate comparison of CoDE and baseline models on test environments with increasing difficulty levels. From left to right, columns correspond to increasing difficulty. From top to bottom, rows correspond to different test environments.

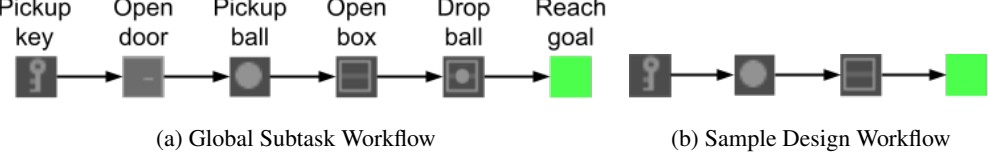

(a) Global Subtask Workflow          (b) Sample Design Workflow

Figure 12: The global subtask workflow (a) and a sample design (b). Sample design respects the dependency structure in the global subtask workflow.

## A.12    List of Test Environments

In Figure 15, we present screenshots of the testing environments with the hardest difficulty levels. While "Login", "Enter Address", "Enter Payment", and "Flight Booking" are single page environments, "Shopping" is a multi-page environment where an agent needs to first navigate the home page and then solve "Login" and "Enter Address" tasks.

## A.13    Example web page designs

In Figure 16, we present more screenshots of generated pages by the adversary from including multi-page websites. They cover a very broad spectrum of complexities and DOM tree structures. As an example, two web pages on the top right both have "City", "CVV", and "Address" elements but with different orders. This allows the web navigation agents to observe a website in multiple different ways for better generalization.

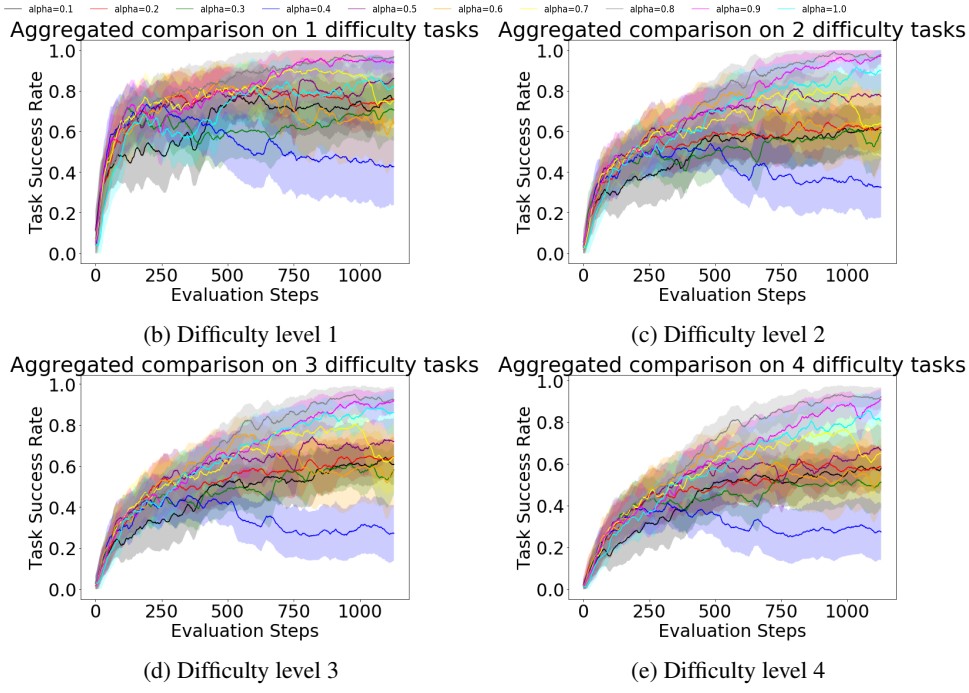

(b) Difficulty level 1            (c) Difficulty level 2

(d) Difficulty level 3            (e) Difficulty level 4

Figure 13: Aggregated task success rate comparison of CoDE trained with different $\alpha$ weights. CoDE yields the best performance with a strong $\alpha$ illustrating the importance of the introduced budget mechanism for the compositional task design in web navigation.

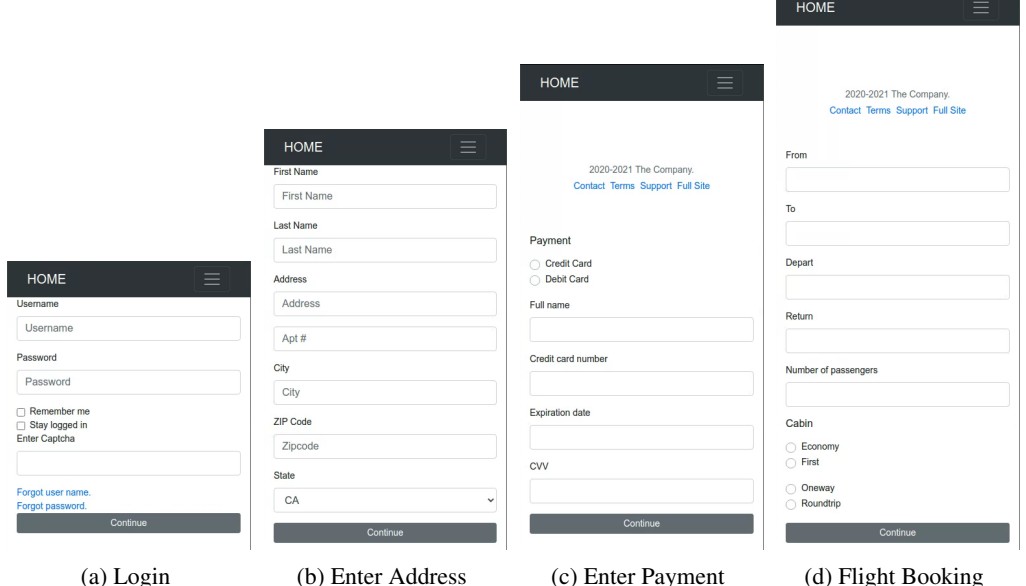

(a) Login      (b) Enter Address      (c) Enter Payment      (d) Flight Booking

Figure 14: Screenshots of single page test environments.

## A.14 Implementation Details on Web navigation and adversary networks

Following [17], we design web navigation agent networks as DOM and profile encoders with pairwise similarity scoring. Each web navigation agent policy network has 104501 parameters.

In Figure 17, we detail the adversary network architecture for a single design action with the parameters used in this work. We use 100 dimensions for hidden vectors for all dense layers as well as the LSTM network. Every dense layer is stacked twice and tanh activation function is applied on

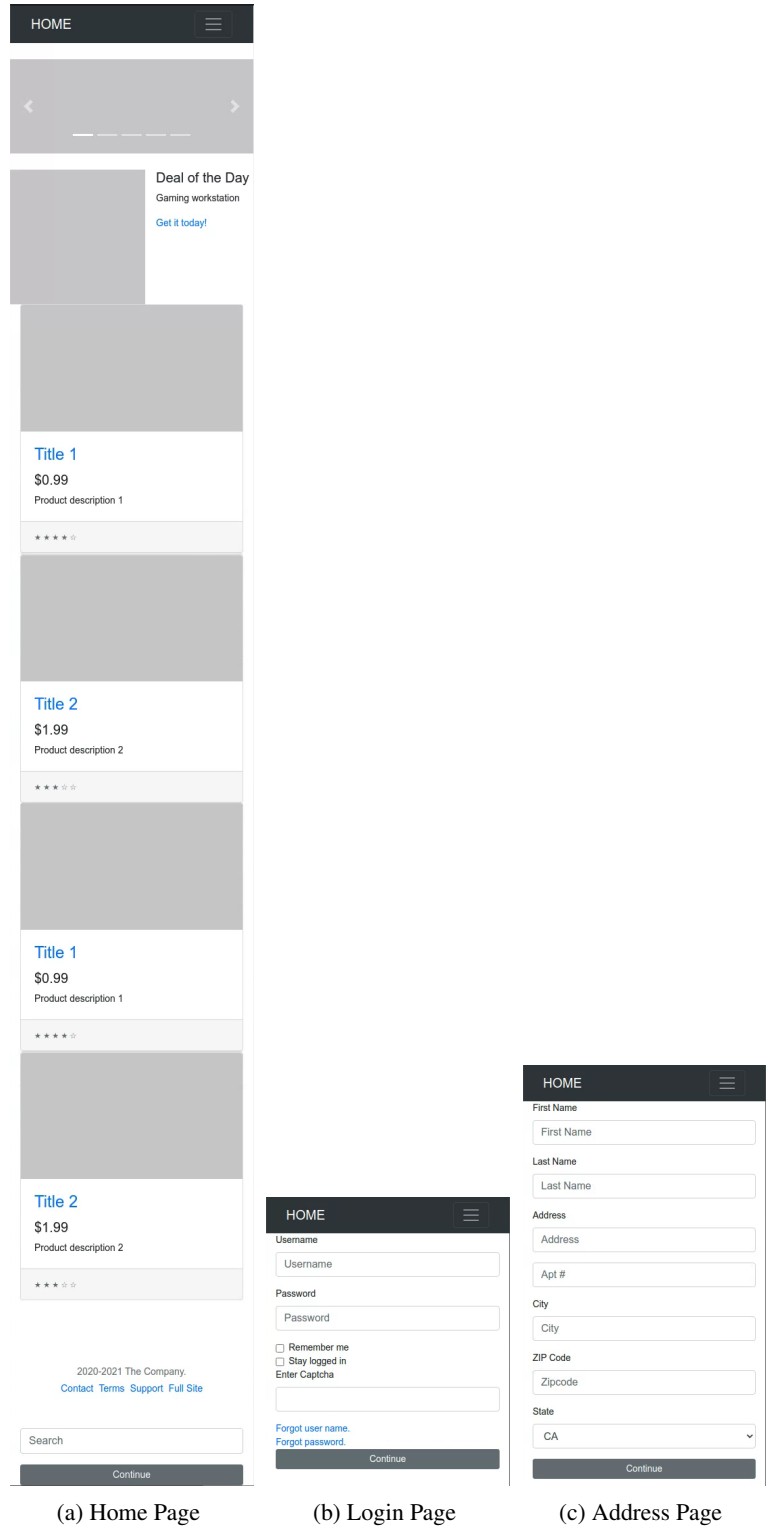

(a) Home Page    (b) Login Page    (c) Address Page

Figure 15: Screenshots of multi-page "Shopping" environment. The "Shopping" environment is composed of a complex home page and additional "Login" and "Enter Address" pages.

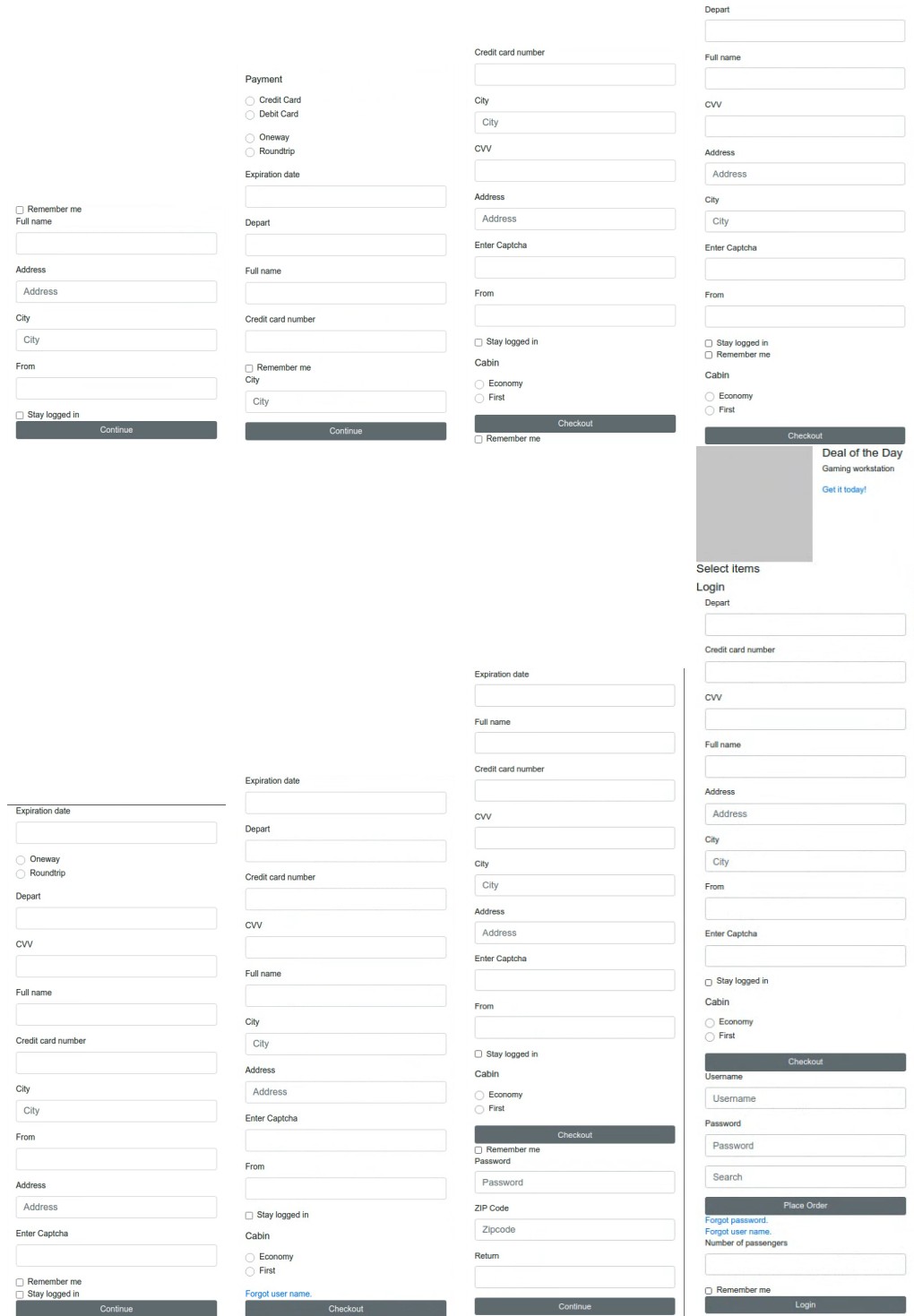

Figure 16: Screenshots of sample pages generated by the adversary.

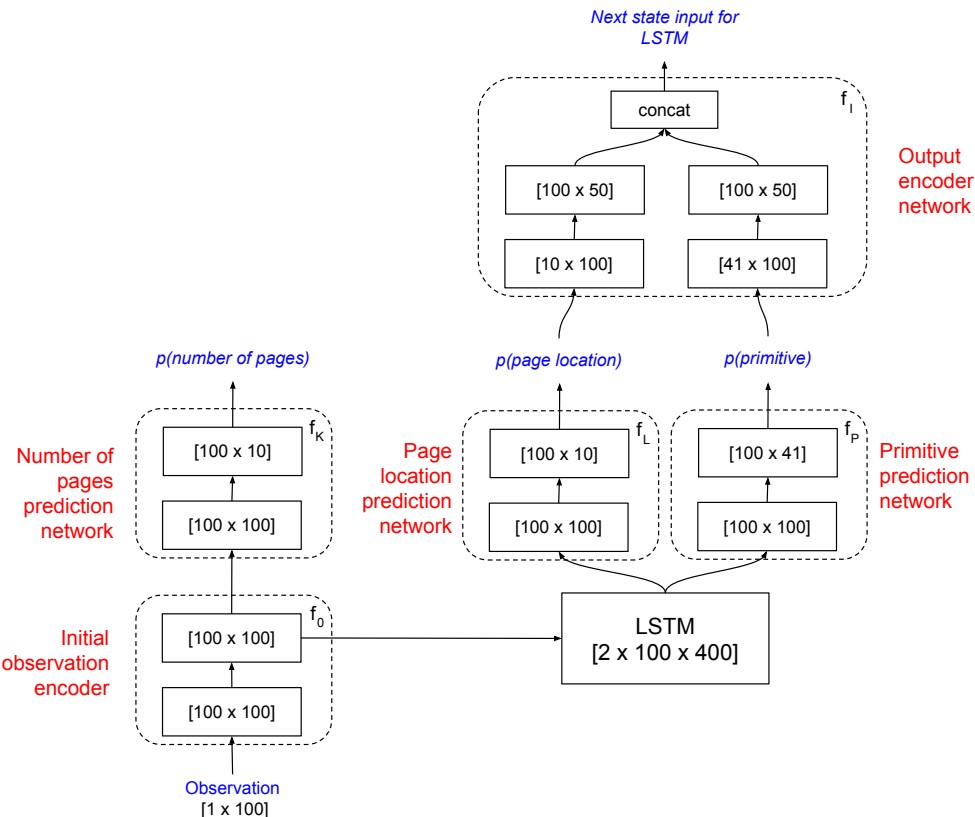

Figure 17: Adversary network architecture. Each box corresponds to a dense layer with the shape of the corresponding linear transformation matrix. Each dense layer also includes a bias vector with the same size as the columns of the corresponding matrices.

the output of all non-final dense layers. Total number of parameters for the adversary policy network is 152461.

For both cMiniGrid and gMiniWoB tasks, we use $\beta = \delta = 0.0$ for reward thresholds and $M = 2$ based on a hyper-parameter search (see Appendix A.3).

### A.15   Limitations

One limitation of our method is that the difficulty objective in Eq. 2 is based on the number of primitives $N$ added to the compositional task. It may not always be the case that difficulty is a direct function of the number of primitives, and in that case the difficulty objective would not generalize to those types of tasks. As an alternative incentive, we provide the population-based regret (PopRegret). Although the empirical results show that optimizing for PopRegret alone learns much more slowly, it is more environment-agnostic and could be useful for a range of tasks.

A second limitation of our method is that it still requires building an environment-specific rendering function that can translate the generator's actions into a feasible compositional task (e.g. by linking web pages together). We rely on pre-defined primitives for each environment to enable the adversary to construct the tasks.

### A.16   Broader Impact

The immediate application of this work is to train RL agents to complete web navigation tasks for users. This could free up time for people, because instead of manually clicking through websites to perform tasks such as flight booking, they could simply issue a language query such as "Book me a flight to San Francisco on Tuesday". We do not foresee that this application would lead to job automation, since by its nature web-based form-filling is a task that does not require interacting with a person. More broadly, enabling RL agents to better perform compositional tasks could be a

step towards future applications such as autonomous driving and household robotics. Each of these applications comes with both potential harms (such as job automation) and benefits (such as increased efficiency, or improved elder care).