# OpenReview forum: "Environment Generation for Zero-Shot Compositional Reinforcement Learning"
_NeurIPS.cc/2021/Conference — NeurIPS 2021 Poster_

### Official Review · Reviewer_w21P · 2021-07-16

**Rating:** 6
**Confidence:** 3

**Summary:**

The paper presents a method, Compositional Design of Environments (CoDE), for automatically constructing a curriculum over a complex domain of environments. The algorithm is framed as a multi-agent game: a generator constructs an environment using a sequence of actions, and afterwards a population of agents attempts to solve the constructed environment. Agents are trained to maximize the extrinsic reward they receive when attempting to solve the constructed environments. The generator reward, the main contribution of this paper, has two components: the first is a population-based regret computed as the difference between the average reward of the best performing agent in the population and the average reward across all agents in the population. The second component is a "difficulty" reward that encourages the generator agent to only complexify its constructed environments so long as one agent in the population remains capable of solving the task to a sufficient degree (given by a success threshold). If all agents fail the environment by another degree (the failure threshold), then the generator receives a negative reward, encouraging it to simplify its environment distribution when necessary.

**Limitations And Societal Impact:**

Limitations are discussed in section 8 in the main text and in the appendix.

A potentially major limitation of this work is that it requires both a way to represent tasks efficiently so that a generator can produce a valid task description in a small number of actions, as well as a way to transform simple task descriptions into environments that can be executed by reinforcement learning agents. Constructing these components could conceivably be more expensive in some domains than hand-designing a curriculum of training tasks.


**Main Review:**

The paper presents an interesting solution to automated curriculum design where the environments are themselves constructed by a reinforcement learning agent. The paper is most similar to previous work PAIRED, which also used regret as a learning signal for an environment generator agent. However, the main novelty presented in the paper is (1) a better regret estimate that uses a population of agents rather than only 2 (protagonist / antagonist) and (2) a difficulty reward signal that provides denser learning signal then regret only. Both improvements make sense intuitively and are clearly motivated in the main text.

The results and ablations seem to support that both of these modifications provide significant improvements over the PAIRED baseline. The results on gMiniWoB and the real website evaluation seem especially significant, and the real website dataset could be an interesting benchmark by itself for future research.

However, there are some confusing aspects in the experiment section:
1) On L295-296, it claims that CoDE only uses a population size of 2 for fair comparison to PAIRED. Can this then be understood that the CoDE results in the experiments section are PAIRED but with the added difficulty reward?
2) Given that information, what is the difference between PopRegret and CoDE in e.g. Figure 6?
3) How does population size affect final performance? This ablation seems important to motivating using |P| > 2.
4) There is a mention of "Flexible PAIRED" and "budget enforcement" in Figure 6, but no additional information. Can you explain what is meant here?
5) The success/failure thresholds of the difficulty reward seem like potentially important hyperparameters that can largely affect the final results of CoDE. I can't easily find what values these were set to in the text, can you describe this in more detail in the experiments section, including how you ended up at the values you chose?
6) It would be interesting to see an ablation of these thresholds, and the performance of CoDE for various values. This would determine how critical these thresholds are to performance, and how easy they are to tune for agent designers.
7) If the reward of an agent is unbounded, would using static thresholds still make sense? Or would you also need to increase the threshold with increasing/decreasing agent returns?
8) It's mentioned that agents are optimized using a policy gradient method. Given the highly non-stationary environment that the population is trained under, was there any observation that agent optimization is affected by the shifts in environment distributions over training (e.g. catastrophic forgetting or learning speed reductions due to activations saturating during training)?
9) In Figure 11, is there any intuition on why the CL baseline increases rapidly in performance but eventually collapses?

Additionally, can the authors comment on whether it is possible the generator and agent population fall into some unwanted local minima and how these solutions can be avoided? I think there is the possibility on very complex domains that the learning process gets stuck in a part of the game which might not overlap with the test distribution, e.g. in the space of creating any website, the generator can start designing extremely complex websites where even humans can not easily solve them.

In conclusion, I believe that the method is well-motivated and sufficiently empirically validated, and remains a useful contribution to the field of automated environment generation despite there being some limitations.

**Time Spent Reviewing:**

4

---

> ### Author Response · Authors · 2021-08-06
> **Detailed response to comments**
>
> Thank you for your insightful and detailed feedback. We will address each of your comments below.
>
> > On L295-296, it claims that CoDE only uses a population size of 2 for fair comparison to PAIRED. Can this then be understood that the CoDE results in the experiments section are PAIRED but with the added difficulty reward?
>
> No, even with a population of 2, population-based regret is distinct from PAIRED. PAIRED uses a fixed antagonist agent, which can cause the training to stall or fail to converge (see the discussion of “Lack of convergence and stalled training” in lines 175-181 of the paper). In PopRegret, the regret is computed with respect to the agent that is currently performing best, which can flexibly switch each episode. This addresses the stalled training problem present in PAIRED, which can be observed to occur in the Shopping and Flight Booking tasks in Figure 6.
>
> > what is the difference between PopRegret and CoDE in e.g. Figure 6?
>
> PopRegret does not include the difficulty budget of Equation 2.
>
> > How does population size affect final performance? This ablation seems important to motivating using |P| > 2.
>
> We initially experimented with a larger population of agents, but we found it did not significantly increase performance, but did come at an additional computational cost. Therefore, we chose to use N=2 in our experiments to ensure a fair comparison with PAIRED, since otherwise the performance increase of our algorithm could be attributed to the additional computation time it used.
>
> > There is a mention of "Flexible PAIRED" and "budget enforcement" in Figure 6, but no additional information. Can you explain what is meant here?
>
> We apologize, this is incorrect terminology and we will correct this in the revision. Here “Flexible PAIRED” refers to PopRegret and Flexible PAIRED with budget enforcing is equivalent to the full CoDE algorithm.
>
> > The success/failure thresholds of the difficulty reward seem like potentially important hyperparameters that can largely affect the final results of CoDE. I can't easily find what values these were set to in the text, can you describe this in more detail in the experiments section, including how you ended up at the values you chose?
>
> In the environments studied here, setting the success/failure thresholds is straightforward. We set $\beta=\delta=0$, because finishing any subtask leads to a positive reward, and making mistakes leads to a negative reward. We will clarify this in the paper text.
>
> > It would be interesting to see an ablation of these thresholds, and the performance of CoDE for various values. This would determine how critical these thresholds are to performance, and how easy they are to tune for agent designers.
>
> Although success/failure has an intuitive meaning in compositional tasks, it should be easy to set a similar threshold for any task by normalizing the reward such that the mean reward is 0. Then, setting $\beta=\delta=0$ would encourage the adversary to increase the difficulty when the agents achieve above-average performance, and decrease the difficulty for below-average performance.
>
> > If the reward of an agent is unbounded, would using static thresholds still make sense? Or would you also need to increase the threshold with increasing/decreasing agent returns?
>
> As mentioned above, normalizing the rewards would enable coping with the agent’s performance increasing over time.
>
> > It's mentioned that agents are optimized using a policy gradient method. Given the highly non-stationary environment that the population is trained under, was there any observation that agent optimization is affected by the shifts in environment distributions over training (e.g. catastrophic forgetting or learning speed reductions due to activations saturating during training)?
>
> Great question. We believe that policy-gradient-based methods may be better suited to the type of non-stationary, multi-agent training scenario described in this paper. Recent work (https://arxiv.org/abs/2011.09533, and https://arxiv.org/abs/2103.01955) has shown that policy-gradients, and in particular PPO, are more effective than state-of-the-art off-policy multi-agent methods for a range of multi-agent scenarios. It is hypothesized that the stable updates of PPO mean that the other agents cannot suddenly shift their policy and therefore induce drastic shifts in the distribution of environments over training. Given that our generator agent must adapt to changing policies of the learner agents, we expect that PPO will give the most stable performance.
>
> > In Figure 11, is there any intuition on why the CL baseline increases rapidly in performance but eventually collapses?
>
> Yes, the reason is that the scheduled curriculum learning approach continues to add primitives even after the problem becomes too difficult for the agents to solve, therefore impairing their ability to learn. We will add an explanation of this result to the text.
>
> > Additionally, can the authors comment on whether it is possible the generator and agent population fall into some unwanted local minima and how these solutions can be avoided? I think there is the possibility on very complex domains that the learning process gets stuck in a part of the game which might not overlap with the test distribution, e.g. in the space of creating any website, the generator can start designing extremely complex websites where even humans can not easily solve them.
>
> Great question. In theory, for a sufficiently large task distribution there may only be a subset of the space that humans are actually interested in. In this case it would be worth constraining the generator to focus on the part of the distribution that resembles real/useful tasks. In the web navigation domain, this could be performed by evaluating the log likelihood of the website under a generative model trained on real websites. However, in the case where there are potentially extremely *complex* tasks that humans cannot solve, we believe our algorithm would not focus on those tasks until the agents have attained proficiency on the less complex, solvable tasks. This is because the algorithm is designed to produce a curriculum that focuses on learning easy tasks first.
>
> > A potentially major limitation of this work is that it requires both a way to represent tasks efficiently so that a generator can produce a valid task description in a small number of actions, as well as a way to transform simple task descriptions into environments that can be executed by reinforcement learning agents. Constructing these components could conceivably be more expensive in some domains than hand-designing a curriculum of training tasks.
>
> We acknowledge that parameterizing the tasks/generator required engineering effort, but we believe our method would still lead to better coverage of the underlying task distribution than manually constructing a curriculum of example tasks. The reason is that the compositional primitives can be combined in an exponential number of ways to form a vast array of tasks, and manually designing an exponential number of tasks by hand would be prohibitively expensive.
>
> > the main novelty presented in the paper is (1) a better regret estimate that uses a population of agents rather than only 2 (protagonist / antagonist) and (2) a difficulty reward signal that provides denser learning signal then regret only.
>
> Thank you for recognizing our contributions of the regret and difficulty objectives. We would like to respectfully point out the additional contributions of (3) formalizing the space of compositional tasks with Petri Nets, (4) building open-source benchmarks which allow parameterizing and generating a diverse range of compositional tasks, and (5) applying environment-generation curriculum methods to learning compositional tasks for what we believe is the first time.

---

> > ### Author Response · Authors · 2021-08-11
> > **New experimental results with ablation studies**
> >
> > We wanted to give an update with our new results on different ablation experiments. We evaluated on tasks with difficulty=4. We also added a “number of evaluation steps” column for experiments that are still running so that they are comparable to the results in our paper. We will put final results into our paper as well.
> >
> > **Experimental Setting**
> >
> > *Ablations:* We take beta=delta and experiment with a negative (beta=-0.2) and a positive (beta=0.2) reward threshold. We experimented with using M=4 or M=6 rollouts per training iteration. We use different budget weights (budget=0.8 and budget=0.9) for each of the ablations as well.
> >
> > **Analysis**
> >
> > Beta is an important hyperparameter that needs careful tuning for the best performance but CoDE still outperforms baselines for different values. We observe that using a relatively high reward threshold (0.2) causes the adversary to become more conservative and increase complexity only when navigators are performing very strongly.
> >
> > Using a larger number of episodes can decrease performance. Note that these experiments are still running.
> >
> > | Experiment | Login | Address | Payment | Flight | Shopping | Number of evaluation steps |
> > | :----------- | :-----------: | :-----------: | :-----------: | :-----------: | :-----------: | :-----------: |
> > CoDE    (Paper)     | 92%   |    96%   |     96% | 83% | 93% | 1150 |
> > beta=-0.2,budget=0.8 | 86% | 99% | 90% | 96% | 87% | 1150 |
> > beta=0.2,budget=0.8 | 53% | 23% | 37% | 27% | 7% | 800 |
> > beta=-0.2,budget=0.9 | 82% | 94% | 82% | 88% | 70% | 1150 |
> > beta=0.2,budget=0.9 | 48% | 35% | 37% | 29% | 15% | 800 |
> > M=4,budget=0.8 | 82% | 64% | 74% | 61% | 44% | 600 |
> > M=4,budget=0.9 | 84% | 81% | 80% | 81% | 79% | 600 |
> > M=6,budget=0.8 | 63% | 43% | 42% | 33% | 2% | 400 |
> > M=6,budget=0.9 | 53% | 46% | 52% | 56% | 2% | 600 |

---

### Official Review · Reviewer_dK3z · 2021-07-17

**Rating:** 6
**Confidence:** 3

**Summary:**

This paper proposes a framework to generate compositional tasks as automated curricula using a graph representation. The proposed method jointly trains a task generator and the RL agent by maximizing the regret between the trained agents and the best-performing agent and the difficulty of the environment. Experiments are conducted in two domains: MiniGrid and MiniWob.

**Limitations And Societal Impact:**

The authors provide discussions on the limitation of applying the difficulty budget to tasks without a goal state. It would be even better if the authors could suggest alternative solutions for such scenarios.

**Main Review:**

Strengths:

- Compared to prior work, the use of Petri Net graph formalism and the formulation of regret and difficulty are original in this paper. The proposed CoDE algorithm has enough novelty.

- The proposed graph formalism and the CoDE algorithm seem to be reasonable. Detailed ablation studies are provided to analyze the performance of the proposed algorithm. However, a few important baselines are missing in the experiments and there is no ablation study to analyze the design choices in the CoDE algorithm.

- The paper is mostly well written and easy to follow.

- This paper address an important problem of generating tasks as curricula and proposes an interesting solution. However, the significance of the proposed method was not fully justified due to missing comparisons with several state-of-the-art methods.

Weaknesses:

- One of my major concerns is that many state-of-the-art curriculum learning methods are not compared in the experiments, including ALP-GMM (Portelas et al. CoRL 2019), GoalGAN (Florensa et al. ICML 2018), and VDS (Zhang et al. NeurIPS 2020). Without comparisons with these recent works, it would be hard to justify the effectiveness of the proposed method.

- The state/action spaces and the parameterization of the environments are not clearly explained in Sec. 6. Such information is important for understanding the complexity of the problems and should be mentioned in the paper.

- The analysis in Sec. 4 is really helpful for the readers to better understand the problem. However, it would be better if more experimental results are provided to prove the hypothesis.

- (Minor) The pargraph of the generator model in Sec. 5 is a bit confusing since some of the descriptions are specific to gMiniWob domain (e.g. "page index" in line 231). It would be better to describe the model design in a domain-agnostic manner.

- Some of the citations on curriculum learning and task generation are missing:

    -- M. Svetlik, Matteo Leonetti, J. Sinapov, Rishi Shah, Nick Walker, and P. Stone. Automatic curriculum graph generation for reinforcement learning agents. In AAAI Conference on Artificial Intelligence, 2017.

    -- B. Peng, J. MacGlashan, R. Loftin, M. Littman, D. Roberts, and Matthew E. Taylor. Curriculum design for machine learners in sequential decision tasks. IEEE Transactions on Emerging Topics in Computational Intelligence, 2:268–277, 2018.

    -- K. Fang, Y. Zhu, A. Garg, A. Kuryenkov, V. Mehta, L. Fei-Fei, and S. Savarese. Learning task-oriented grasping for tool manipulation from simulated self-supervision. Robotics: Science and Systems, 2018.

    -- Wojciech Czarnecki, Siddhant M. Jayakumar, Max Jaderberg, Leonard Hasenclever, Yee Whye Teh, Nicolas Manfred Otto Heess, Simon Osindero, and Razvan Pascanu. Mix&match - agent curricula for reinforcement learning. In International Conference on Machine Learning, 2018.

**Time Spent Reviewing:**

3 hours

---

> ### Author Response · Authors · 2021-08-06
> **New experiments and clarifications regarding prior work**
>
> Thank you for your feedback and insights into the paper. We will include the suggested references, and incorporate your other suggestions, including to remove “page index” from Section 5 to make the section domain-agnostic.
>
> Thank you for your point about related methods such as ALP-GMM (Portelas et al. CoRL 2019), GoalGAN (Florensa et al. ICML 2018), and VDS (Zhang et al. NeurIPS 2020). Our paper does already include citations to these works, but we did not benchmark against them because the aims of these papers are distinct from the problem we consider. These methods do not address the problem of learning how to perform compositional tasks, nor actually building new environments out of a set of primitives in order to form a curriculum. However, based on your feedback, we are initiating experiments to incorporate a modified version of ALP as an additional baseline. We further address the feasibility and relevance of each baseline below:
> - ALP-GMM focuses on the setting where the environment is parameterized by several independent, continuous parameters, which can then be learned via a multi-armed bandit. While it is not clear how to generate web sites with such a multi-armed bandit approach, the idea of using a similar learning progress objective could be applied here. We have begun experiments to use learning progress as an objective, and will update you with the results.
> - GoalGAN relies on generating an image of a goal state that the agent then learns to reach. In our web navigation setting, given a particular environment the agent’s goal state is always the same; the information must be filled into the web form in exactly the same way. Instead, each episode we must actually generate an entirely new web site for the agent to navigate, which cannot easily be represented as a single image. GoalGAN also requires the goal to be a part of the state space. That is not suitable for web navigation as (i) we specify the goal as a profile which is independent of the website and (ii) we cannot generate a single state (a single DOM) from a complex website or even multiple DOMs. Instead, our method allows us to be agnostic to the representation of the observations and states; in web navigation the observations are DOM elements, and in the maze environment the observations are images. We also note that GoalGAN uses a manual difficulty threshold (GOID) which is similar to our difficulty objective, but we also introduce the population-based regret objective, which does not require environment-specific thresholds.
> - VDS does not generate new environments/tasks, but chooses which task to present next out of a set of pre-defined available tasks. For web navigation, we could use the pre-defined available tasks in the MiniWoB benchmark as the set of tasks to prioritize. However, we show in Table 1 that MiniWoB can already be solved with a scheduled curriculum learning approach, so VDS is unnecessary. Instead, we could start with all possible 10^10 gMiniWob tasks, but since VDS has no mechanism for searching over the available tasks without evaluating the ensemble on each one, it would be prohibitively expensive. In contrast, our method learns how to construct tasks to optimize for the regret and difficulty objectives, thus pursuing a more constrained search through the space of available tasks.
>
> We will include the discussion of these differences in the revised version of the paper.
>
> Rather than benchmark against these works, we have included four competitive baselines based on the state-of-the-art in training web navigation agents ([20,34,16] in our paper), and a state-of-the-art method for generating environments based on minimax regret [7]. These baselines were chosen both because they give competitive performance, and they are focused on a similar setting to our work.
>
> Thank you for your suggestion to include ablation studies of the parameters M, beta, and delta. We are initiating an ablation study of the effect of M, and will update you with the results. Regarding beta and delta, we set beta=delta=0 in our experiments. We believe this is a relatively flexible way to set thresholds regardless of environment, because it is always possible to normalize the rewards such that the mean reward is 0, and thus the threshold is the mean reward value.
>
> Regarding the parameterization of the environments: Section 5 explains how the generator model is parameterized by having two sets of actions, $a \in A$ which allows it to select one of the available primitives, and $b \in (1…K)$ which places the primitive into one of the available subtasks (lines 223-241 and Equation 3). The actual primitives for both environments are described in the Supplementary Material; Algorithm 2 in Section A.6 explains how we extract primitives from a Document Object Model (DOM template), and Section A.8 describes the primitives available in the MiniGrid domain:  (i) Pick up the key, (ii) Open the door, (iii) Pick up the ball, (iv) Open the box, and (v) Drop the ball. We will move some of these details to the main text to make this more clear.

---

> > ### Author Response · Authors · 2021-08-11
> > **New Experimental Results with ALP baseline and ablation studies**
> >
> > We wanted to give an update with our new results to address your concerns. We ran experiments with an ALP-based [1] reward for the adversary and ablation experiments for different hyper parameters. We evaluated on tasks with difficulty=4. We also added a “number of evaluation steps” column for experiments that are still running so that they are comparable to the results in our paper. We will put final results into our paper as well.
> >
> > **Experimental Setting**
> >
> > *ALP:* Similar to ALP-GMM, we use the absolute learning progress which is the absolute difference between rewards of the navigator at timestep “t” and “t-1” as the reward for the adversary. We didn’t add any budget enforcing objectives and use only a single navigator.
> >
> > *Ablations:* We take beta=delta and experiment with a negative (beta=-0.2) and a positive (beta=0.2) reward threshold. We experimented with using M=4 or M=6 rollouts per training iteration. We use different budget weights (budget=0.8 and budget=0.9) for each of the ablations as well.
> >
> > **Analysis**
> >
> > CoDE outperforms ALP-based baseline by a large margin. ALP-based reward exhibits a similar behavior as the PAIRED algorithm where if rewards at consecutive timesteps are very similar, adversary reward becomes miniscule and non informative.
> >
> > Beta is an important hyperparameter that needs careful tuning for the best performance but CoDE still outperforms baselines for different values. We observe that using a relatively high reward threshold (0.2) causes the adversary to become more conservative and increase complexity only when navigators are performing very strongly.
> >
> > Using a larger number of episodes can decrease performance. Note that these experiments are still running.
> >
> > | Experiment | Login | Address | Payment | Flight | Shopping | Number of evaluation steps |
> > | :----------- | :-----------: | :-----------: | :-----------: | :-----------: | :-----------: | :-----------: |
> > CoDE  (Paper)      | 92%   |    96%   |     96% | 83% | 93% | 1150 |
> > ALP | 23% | 15% | 7% | 6% | 2% | 1150 |
> > beta=-0.2,budget=0.8 | 86% | 99% | 90% | 96% | 87% | 1150 |
> > beta=0.2,budget=0.8 | 53% | 23% | 37% | 27% | 7% | 800 |
> > beta=-0.2,budget=0.9 | 82% | 94% | 82% | 88% | 70% | 1150 |
> > beta=0.2,budget=0.9 | 48% | 35% | 37% | 29% | 15% | 800 |
> > M=4,budget=0.8 | 82% | 64% | 74% | 61% | 44% | 600 |
> > M=4,budget=0.9 | 84% | 81% | 80% | 81% | 79% | 600 |
> > M=6,budget=0.8 | 63% | 43% | 42% | 33% | 2% | 400 |
> > M=6,budget=0.9 | 53% | 46% | 52% | 56% | 2% | 600 |
> >
> > [1] Rémy Portelas, Cédric Colas, Katja Hofmann, Pierre-Yves Oudeyer, Teacher algorithms for curriculum learning of Deep RL in continuously parameterized environments, CoRL'19

---

> > > ### Comment · Reviewer_dK3z · 2021-08-19
> > > **Response to the Authors**
> > >
> > > I appreciate the additional experiments (comparisons with ALP and ablations on the hyperparameters) the detailed analysis. Thus I've raised my rating to 6.
> > >
> > > Could you please also include the details of the state/action spaces and the parameterization of the environments?

---

> > > > ### Author Response · Authors · 2021-08-25
> > > > **State/Action Spaces and Parameterization of Environments**
> > > >
> > > > We are glad that our new results are well received and thank your for this positive feedback!
> > > >
> > > > In our **web navigation** experiments, we use a maximum of 250 elements in a give page, 25 profile fields, 40 primitives, and 10 pages.
> > > >
> > > > For *web navigator* agents, input space is 250\*25=6250 pairs of elements and profile fields and action space is 40\*25=1000 pairs of profile fields and primitives.
> > > > For the *adversary*, we use a 100 dimensional random input observation and the output is 40\*10=400 pairs of primitives and pages.
> > > > Our web environments are parameterized by a list of (primitive, page) pairs where primitives are appended to their corresponding pages in the order they are received.
> > > > Please also note that different permutations of elements and primitives would grow state/action spaces.
> > > >
> > > > In our **grid** experiments, we use 100 dimensional random inputs and 5 output primitives for the adversary. We use 8x8=64 dimensional image observations and 6 actions for the navigator agents. Our grid environments are parameterized by a set of primitives (or subtasks).
> > > >
> > > > We will also include these details in our paper.

---

> ### Author Response · Authors · 2021-08-16
> **Follow up**
>
> We wanted to follow up with you on if your concerns have been addressed. Please let us know if there are any remaining concerns, we would be happy to address them.

---

### Official Review · Reviewer_uPhg · 2021-07-19

**Rating:** 7
**Confidence:** 3

**Summary:**

The paper addresses the problem that RL is bad at learning the compositional rules guiding compositional problems. It proposes a two pronged approach: first, it proposes a task (de)composition strategy that allows building up tasks (or POMDPs) compositionally. Second, it proposes a regret-driven curriculum strategy that is able to construct compositional tasks of increasing complexity without supervision, which, in turn, can be used to train a learner that is then able to pick up on the rules of compositionality guiding the domain.

**Main Review:**

Pros:
- The paper is very well written and easy to follow.
- The goal of the paper is extremely important: very many real-world scenarios are compositional, and RL is severely lacking in this regard.
- The individual design choices are well motivated.
- The core idea, guiding the complexity of the domains given to the learning agent by a measure of regret, is intuitively plausible.
- The authors test on existing domains (which appear to be solved) and design new harder domains that they release as open source.

Cons:
- The first contribution of the paper, the hierarchical and compositional decomposition of tasks using petrinets, is only explained in the appendix. Given the relevance to the overall goals of the paper, I think that at least a paragraph or two should be spend on that in the main paper.
- There is a discrepancy in the complexity of the two tasks: it looks as if the website domain is vastly more complex than the gridworld domain. Given that the gridworld domain would be more general and thus more relevant to the research community, it would have been nice to have this also scale to high complexity.
- I find myself doubting the value of the new domains the authors are releasing along with the paper. Not because I am not convinced of their relevance (in fact, I do believe that the domains could be highly relevant), but because the suggested approach in the paper solves these domains (success > 95%), effectively making them redundant before they were even released to the public.

Originality:
To my knowledge, the proposed regret-based curriculum training strategy is novel and a creative solution to a fundamental problem.

Quality:
The quality of the submission is very high. The language is clear, the paper is well structured, and the illustrations are good (though the text on the illustrations could be larger and sharper).

Clarity:
The clarity of the submission is high. Each argument made is easy to follow, and the overall structure makes it easy to understand each point in context.

Significance:
The core problem of the paper - RL's difficulties in generalizing over compositional domains - is highly significant. The significance of the entire paper is somewhat reduced by their choice of narrow domain.

**Time Spent Reviewing:**

3

---

> ### Author Response · Authors · 2021-08-06
> **Thank you and clarifications regarding gMiniWoB**
>
> Thank you for your comments; we appreciate the detailed feedback and will incorporate your suggestions. In particular, we will expand the discussion of the Petri Nets formalism by moving some of the content of Supplementary Material section A.1 into the main text.
>
> We would like to clarify an important point about the open-source web navigation domain gMiniWoB. gMiniWob consists of both the framework and code to generate websites out of primitives, and a set of evaluation environments. You are correct that our agent reaches 95% on the gMiniWoB evaluation environments. We note that these evaluation sites are still a significantly harder benchmark than the prior MiniWoB benchmark, as shown by the results in Table 1 (see especially the performance of CL in both), and the fact that prior state-of-the-art methods reach only 0-20%. However, gMiniWob can be easily extended to include harder evaluation websites. Based on your suggestions, we will design more challenging evaluation websites and update the open-source release of gMiniWob to include them.
>
> Regardless of the evaluation component, gMiniWoB still provides a general framework for designing a large array of training websites, and was a key component in enabling our algorithm to reach state-of-the-art performance on real websites.
>
> Here are some features of the web navigation domain that we think show it has significance and is not too narrow. First, agents that could successfully navigate websites to complete tasks such as purchasing flights or purchasing office supplies could provide significant economic value. Second, it is a challenging and open problem, and production-quality approaches that can reliably navigate real-world websites do not currently exist. As you can see from Table 2, our best methods achieve only 12-62% on real websites. There is still a great deal of research and development required before RL agents can generally solve real web navigation tasks. Therefore, web navigation represents a valuable, challenging, as-yet-unsolved, real-world example of a compositional task.

---

> > ### Comment · Reviewer_uPhg · 2021-08-19
> > **Not fully convinced**
> >
> > I remain unconvinced that the new domain is of great value for the community, since I have not seen a more complex version where the proposed method fails.
> >
> > However, I greatly look forward to a more complex version being released soon, which I would possible use and cite.
> >
> > Thus, I stick with my "accept" score of 7.

---

> > > ### Author Response · Authors · 2021-08-25
> > > **A More Complex Benchmark**
> > >
> > > Thank you for this valuable and encouraging feedback!
> > >
> > > We will work on adding more complex environments with more domains to the open source benchmark that we will release. We would like to also remind that using a much larger real primitive database in our framework, we see a bigger challenge and a drop in performance (Table 2). We believe our framework would help promote further research and benchmarks in compositional RL.

---

### Author Response · Authors · 2021-08-11
**New Experimental Results**

We wanted to share our new results using an ALP-based [1] reward for the adversary and ablation experiments for different hyper parameters. We evaluated on tasks with difficulty=4. We also added a “number of evaluation steps” column for experiments that are still running so that they are comparable to the results in our paper. We will put final results into our paper as well.

**Experimental Setting**

*ALP:* Similar to ALP-GMM, we use the absolute learning progress which is the absolute difference between rewards of the navigator at timestep “t” and “t-1” as the reward for the adversary. We didn’t add any budget enforcing objectives and use only a single navigator.

*Ablations:* We take beta=delta and experiment with a negative (beta=-0.2) and a positive (beta=0.2) reward threshold. We experimented with using M=4 or M=6 rollouts per training iteration. We use different budget weights (budget=0.8 and budget=0.9) for each of the ablations as well.

**Summary**

CoDE outperforms ALP-based baseline by a large margin. Beta is an important hyperparameter that needs careful tuning for the best performance but CoDE still outperforms baselines for different values. Using a larger number of episodes can decrease performance. Note that these experiments are still running.

| Experiment | Login | Address | Payment | Flight | Shopping | Number of evaluation steps |
| :----------- | :-----------: | :-----------: | :-----------: | :-----------: | :-----------: | :-----------: |
CoDE (Paper)         | 92%   |    96%   |     96% | 83% | 93% | 1150 |
ALP | 23% | 15% | 7% | 6% | 2% | 1150 |
beta=-0.2,budget=0.8 | 86% | 99% | 90% | 96% | 87% | 1150 |
beta=0.2,budget=0.8 | 53% | 23% | 37% | 27% | 7% | 800 |
beta=-0.2,budget=0.9 | 82% | 94% | 82% | 88% | 70% | 1150 |
beta=0.2,budget=0.9 | 48% | 35% | 37% | 29% | 15% | 800 |
M=4,budget=0.8 | 82% | 64% | 74% | 61% | 44% | 600 |
M=4,budget=0.9 | 84% | 81% | 80% | 81% | 79% | 600 |
M=6,budget=0.8 | 63% | 43% | 42% | 33% | 2% | 400 |
M=6,budget=0.9 | 53% | 46% | 52% | 56% | 2% | 600 |

[1] Rémy Portelas, Cédric Colas, Katja Hofmann, Pierre-Yves Oudeyer, Teacher algorithms for curriculum learning of Deep RL in continuously parameterized environments, CoRL'19

---

### Public Comment · ~Stefanos_Nikolaidis1 · 2021-11-22
**Interesting work!**

Very interesting work, it is exciting to see sequential environment generation as part of a curriculum for complex compositional tasks!

It'd be interesting to leverage recent methods on generating complex and diverse environments as part of a zero-shot RL framework. For example, in the RSS 2021 paper (https://arxiv.org/pdf/2106.10853.pdf), we automatically generate Overcooked environments that result in diverse coordination behaviors by searching the latent space of a GAN with the QD algorithm CMA-ME and by repairing the generated environments with a mixed integer program, so that the environments are executable by the agents. The fact that all generated environments are executable and they promote large variations in performance could be beneficial in a curriculum setting.

---

### Decision · Program_Chairs · 2021-09-27

**Decision:**

Accept (Poster)

**Comment:**

This paper makes several solid contributions — 1) a method to decompose tasks in a compositional manner, so as to enable RL agents to generalize to unseen, complex test scenarios, and 2) new open-source benchmarks for testing compositional generalization in RL agents, on gridworld and web-based domains. The reviewers appreciated the novelty of ideas presented and agree that the paper is solving an important problem. The authors in their responses promise to add in ablation studies and additional clarifications on the framework (state space, actions, etc.), better comparison to related work on curriculum/self-supervised learning, and adding more complex environments to the open source benchmarks. With these changes, I think this paper will be quite valuable to the community and enable future research on compositional generalization in RL.